

# 1 10–year satellite–constrained fluxes of ammonia improve

# 2 performance of chemistry transport models

**Nikolaos Evangeliou[1,*], Yves Balkanski[2], Sabine Eckhardt[1], Anne Cozic[2], Martin**
**Van Damme[3], Pierre-François Coheur[3], Lieven Clarisse[3], Mark W. Shephard[4],**
**Karen E. Cady-Pereira[5], Didier Hauglustaine[2]**
[1]Norwegian Institute for Air Research (NILU), Department of Atmospheric and Climate
Research (ATMOS), Kjeller, Norway.
[2]Laboratoire des Sciences du Climat et de l'Environnement (LSCE), CEA-CNRS-UVSQ,
91191, Gif-sur-Yvette, France.
[3]Université libre de Bruxelles (ULB), Spectroscopy, Quantum Chemistry and Atmospheric
Remote Sensing (SQUARES), Brussels, Belgium.
[4]Environment and Climate Change Canada, Toronto, Ontario M3H 5T4, Canada.
[5]Atmospheric and Environmental Research, Inc., Lexington, MA, USA.
**\*** Corresponding author: N. Evangeliou (Nikolaos.Evangeliou@nilu.no)





**Abstract**
In recent years, ammonia emissions have been continuously increasing being almost four
times higher than in the 20[th] century. Although an important species as its use as a fertilized
sustains human living, ammonia has major consequences both for humans and the environment,
because of its reactive gas phase chemistry that makes it easily convertible to particles. Despite
its pronounced importance, yet, ammonia emissions are highly uncertain in most emission
inventories. However, the great development of satellite remote sensing nowadays provides the
opportunity for more targeting research in constraining ammonia emissions. Here, we used
satellite measurements to calculate global ammonia emissions over the period 2008–2017.
Then, the calculated ammonia emissions were fed to a chemistry transport model and ammonia
concentrations were simulated for the period 2008–2017.
The simulated concentrations of ammonia were compared with ground measurements
from Europe, North America and Southeastern Asia, as well as with satellite measurements.
The satellite-constrained ammonia emissions represent global concentrations more accurately
than state-of-the-art emissions, which underestimate ammonia with a factor of two. Calculated
fluxes in the North China Plain were increased after 2015, not due to emission changes, but due
to changes in sulfate emissions that resulted in less ammonia neutralization and hence in larger
atmospheric loads. Emissions over Europe were also twice as much as those in traditional
datasets with dominant sources to be industrial and agricultural applications. Four hot-spot
regions of high ammonia emissions were seen in North America characterized by large
agricultural activity (Colorado), animal breeding (Iowa, northern Texas and Kansas), animal
farms (Salt Lake, Cache, and Utah) and animal breeding and agricultural practices (California).
South America is dominated by ammonia emissions from biomass burning, which cause a
strong seasonality. In Southeastern Asia, ammonia emissions from fertilizer plants in China,
Pakistan, India and Indonesia are the most important, while a strong seasonality was observed
with a spring and late summer peak due to rice and wheat cultivation. Modelled concentrations
from the satellite-constrained ammonia emissions are overestimated in Eastern Europe, where
state-of-the-art emissions capture observations better. Measurements of ammonia
concentrations in North America were better reproduced with satellite-constrained emissions,
while all emissions generally underestimate station concentrations in Southeastern Asia. The
calculated ammonia emissions also reproduce global CrIS (Cross-track Infrared Sounder)
observations more effectively.



## 1 Introduction

Ammonia (NH₃) has received a lot of attention nowadays due to its major implications for the population and the environment (Erisman, 2004; Erisman et al., 2007). These include eutrophication of semi-natural ecosystems and acidification of soils (Stevens et al., 2010), secondary formation of particulate matter in the atmosphere (Anderson et al., 2003), and alteration of the global greenhouse balance (De Vries et al., 2011). More specifically in the troposphere, ammonia reacts with the abundant sulfuric and nitric acids (Malm, 2004) contributing 30 % to 50 % of the total aerosol mass of PM2.5 and PM10 (Anderson et al., 2003). Ammonium aerosols are therefore a very important component in regional and global aerosols processes (Xu and Penner, 2012) also having significant implications for human health (Aneja et al., 2009). Ammonia alters human health indirectly mainly through formation of PM2.5 (Gu et al., 2014) that penetrate the human respiratory systems and deposit in the lungs and alveolar regions (Pope III et al., 2002) causing premature mortality (Lelieveld et al., 2015). As regards to the climate impact, the same ammonium aerosol particles affect Earth's radiative balance, both directly by scattering incoming radiation (Henze et al., 2012) and indirectly as cloud condensation nuclei (Abbatt et al., 2006). They may also cause visibility problems and contribute to haze effect due to secondary PM formation.

Sources of ammonia include wild animals (Sutton et al., 2000), ammonia-containing watersheds (Sørensen et al., 2003), traffic (Kean et al., 2009), sewage systems (Reche et al., 2012), humans (Sutton et al., 2000), biomass burning (Sutton et al., 2008) and domestic coal combustion (Fowler et al., 2004), volcanic eruptions (Sutton et al., 2008) and agriculture (Erisman et al., 2007). The latter is responsible for the majority of ammonia global atmospheric emissions. Specifically, in the United States and Europe about 80% of all emissions is related to agriculture (Leip et al., 2015). Emissions have increased considerably since pre-industrial times and are unlikely to decrease due to the growing demand for food and feed (Aneja et al., 2008).

The growing attention in ammonia levels has enabled many monitoring actions in Europe (European Monitoring and Evaluation Programme, EMEP), in Southeastern Asia (East Asia acid deposition NETwork) and in the North America (Ammonia Monitoring Network in the US, AMoN-US; National Air Pollution Surveillance Program (NAPS) sites in Canada) to record surface concentrations of ammonia continuously. Recently, several satellite products have been also developed in an effort to identify global levels of ammonia considering that the




relatively sparse existing monitoring network has an insufficient coverage for this purpose.
These are derived from satellite sounders as the Infrared Atmospheric Sounding Interferometer
(IASI) (Van Damme et al., 2017), the Atmospheric Infrared Sounder (AIRS) (Warner et al.,
2017), the Cross-track Infrared Sounder (CrIS) (Shephard and Cady-Pereira, 2015) the
Tropospheric Emission Spectrometer (TES) (Shephard et al., 2015), and Greenhouse Gases
Observing Satellite (Someya et al., 2020). Both IASI and CrIS ammonia products are being
continuously compared and evaluated against other observations and products. For example,
against column-integrated levels measured by Fourier transform infrared spectroscopy (FTIR)
(Dammers et al., 2016, 2017), ground-based measurements (Van Damme et al., 2015; Kharol
et al., 2018), bottom-up emissions (Van Damme et al., 2018; Dammers et al., 2019) and
atmospheric chemistry transport models (CTMs) (Shephard et al., 2020; Whitburn et al.,
2016a).
Despite its importance, ammonia is a poorly quantified trace gas, with uncertainties over
50% on the global emission budget and even higher on temporal and local scales (Dentener and
Crutzen, 1994; Faulkner and Shaw, 2008; Reis et al., 2009). In the present paper, we grid 10
years (2008–2017) of satellite measurements of ammonia retrieved from IASI to calculate
monthly surface emissions (hereafter named NE) (see section 2). The same is done using the
gridded IASI ammonia column concentrations from Van Damme et al. (2018) (named as VD0.5
and VDgrlf) (see section 2). The three different emission inventories together with a state–of–
the–art one, which is more often used by models (named as EGG), are then imported in a CTM
to simulate ammonia for the same 10–year period. Finally, an evaluation of simulated surface
concentrations against ground–based measurements from different monitoring stations and
satellite products allow to quantify the improvements in ammonia emissions.

## 107    2   Methods

### 108    2.1   Satellite ammonia

### 109    2.1.1   IASI ammonia

The Infrared Atmospheric Sounding Interferometer (IASI) onboard the MetOp-A satellite
measures Earth's infrared radiation twice a day in a spectral range of 645–2,760 cm$^{-1}$ with an
elliptical footprint (at nadir circular with a diameter of 12 km) at nadir (Clerbaux et al., 2009).
Due to the larger thermal conditions that lead to smaller uncertainties, only morning data were
used in the present assessment (Clarisse et al., 2010). The 10–year dataset used here is ANNI-
NH3-v2.1R-I product (Van Damme et al., 2017). The Artificial Neural Network for IASI





(ANNI) algorithm converts the hyperspectral range index to an column–integrated $NH_3$ value
(Whitburn et al., 2016a). The dataset also provides cloud coverage for each measurement
(August et al., 2012). Only measurements with a cloud fraction below 10% were processed in
consistency with Van Damme et al. (2018). Cloud coverage was not provided for all
measurements until March 2010 resulting in smaller data availability before that date. Although
the retrieval algorithm uses a fixed vertical profile, extended validation of the resulting dataset
has verified small uncertainties (Van Damme et al., 2015, 2018; Dammers et al., 2016;
Whitburn et al., 2016b). For instance, Van Damme et al. (2018) reported a difference of
2%±24% (global average) in column–integrated ammonia using different vertical profiles in
the retrieval algorithm.

### 2.1.2  CrIS ammonia

The Cross-Track Infrared Sounder (CrIS) was first launched on the NASA Suomi
National Polar-orbiting Partnership (S-NPP) satellite on 28 October 2011 in a sun-synchronous
low Earth orbit.  The CrIS sensor provides soundings of the atmosphere with a spectral
resolution of 0.625 cm$^{-1}$ (Shephard et al., 2015). One of the main advantages of CrIS is its
improved vertical sensitivity of ammonia closer to the surface due to the low spectral noise of
~0.04K at 280K in the $NH_3$ spectral region (Zavyalov et al., 2013) and the early afternoon
overpass that typically coincides with high thermal contrast, which is optimal for thermal
infrared sensitivity. The CrIS Fast Physical Retrieval (CFPR) (Shephard and Cady-Pereira,
2015) retrieves an ammonia profile (14 levels) using a physics-based optimal estimation
retrieval, which also provides the vertical sensitivity (averaging kernels) and an estimate of the
retrieval errors (error covariance matrices) for each measurement. As peak sensitivity is
typically in the boundary layer between 900 and 700 hPa (~ 1 to 3 km) (Shephard et al., 2020),
the surface and total column concentrations are both highly correlated with the retrieved levels
in the boundary layer.  Shephard et al. (2020) reports estimated total column random
measurement errors of 10–15%, with estimated total random errors of ~30%. The individual
profile retrieval levels have estimated random measurement errors of ~10 to 30 %, with
estimated total random errors increasing to 60 to 100% due to the limited vertical resolution.
These vertical sensitivity and error output parameters are also useful for using CrIS
observations in applications (e.g. data fusion, data assimilation; model-based emission
inversions (e.g., Cao et al., 2020; Li et al., 2019) as a satellite observational operator can be
generated in a robust manner. The detection limit of CrIS measurements has been calculated
down to 0.3–0.5 ppbv (Shephard et al., 2020). CrIS ammonia has been evaluated against other
observations over North America with the Ammonia Monitoring Network (AMoN) (Kharol et



al., 2018) and against ground-based Fourier transform infrared (FTIR) spectroscopy
observations (Dammers et al., 2017) showing small differences and high correlations.

## 2.2  Inverse Distance Weighting (IDW) interpolation

To process large amounts of measurements in a 2-dimensional grid of high resolution,
oversampling methods (Streets et al., 2013) can be used (Van Damme et al., 2018). However,
considering that the resolution of the CTM is 2.5°×1.3° (see section 2.4), there is no need to
process the measurements on such a high-resolution grid and therefore an interpolation method
was used. The method has been extensively used after the Chernobyl accident in 1986 to
process more than 500 thousand deposition measurements over Europe (De Cort et al., 1998;
Evangeliou et al., 2016).
IASI ammonia was interpolated onto a grid of 2.5°×1.3° using a modified Inverse
Distance Weighting (IDW) algorithm described by (Renka, 1988). This method is preferred
due to its ease of use and to its high quality of interpolation. The IDW interpolation is defined
by:

$$\hat{v}(x,y) = \frac{\sum_{i=1}^{n} w_i v_i}{\sum_{i=1}^{n} w_i} \qquad \textbf{Eq. 1}$$

where $\hat{v}(x,y)$ is the interpolated value at point $(x,y)$, $w_1, \ldots, w_i$ are the relative weights and
$v_1, \ldots, v_n$ are the observation values. The weights are defined by the inverse distance functions:

$$w_i = \left(\frac{(r_w - d_i)}{r_w d_i}\right)^2 \qquad \textbf{Eq. 2}$$

$$\text{for } (r_w - d_i) = \begin{cases} r_w - d_i \, if \, d_k < r_w, \\ 0 \, if \, d_k \geq r_w. \end{cases}$$

where $r_w$ denotes the radius of influence of the point $(x_i, y_i)$, $d_i$ the Euclidean distance
between point $(x,y)$ and $(x_i, y_i)$, and $d_k$ is the threshold distance. We used a threshold
distance $(d_k)$ of 50 km, which is similar to the size of each grid cell; different $d_k$ values were
included in a sensitivity study (see section 4.3). The Euclidean distance is calculated using
Vincenty's formulae (Vincenty, 1975).

## 2.3  Emission flux calculation of ammonia

The emission fluxes of ammonia were calculated using a 1–dimensional box model that
assumes first-order loss terms for ammonia and has been already used previously (Van Damme
et al., 2018; Whitburn et al., 2016b). It takes into account the gridded column concentrations of
ammonia that were calculated with the IDW interpolation method and all the potential removal





processes of ammonia occurring in a hypothetical atmospheric box according to the following
equation:
$$E_{NH3} = M_{NH3}/\tau \qquad \text{Eq. 3}$$
where $M_{NH3}$ is the mass of ammonia in each atmospheric box (grid-cell) in molecules cm$^{-2}$ and
$\tau$ is the lifetime of ammonia in the box (given in seconds).
Van Damme et al. (2018) assumed a constant lifetime for ammonia, admitting that this is
a limiting factor of their study on the basis that chemical loss and deposition are highly variable
processes that can change the lifetime drastically. To tackle the large variability of the lifetime
of ammonia, we used gridded lifetime calculated from a CTM. This gives robustness in the
calculated emissions fluxes considering that at regions where sulfuric and nitric acids are
abundant, the chemical loss will be more intense and, thus, lifetime will be much shorter
affecting emissions dramatically.
The lifetime ($\tau$) of ammonia in each grid-box results from the three processes affecting
ammonia concentrations: transport ($t_{trans}$) in and out of the grid-cell, chemical loss ($t_{chem}$)
and deposition ($t_{depo}$):
$$\frac{1}{\tau} = \frac{1}{t_{trans}} + \frac{1}{t_{chem}} + \frac{1}{t_{depo}} \qquad \text{Eq. 4}$$
In a CTM, the lifetime can be easily calculated from the species mass balance equation (Croft
et al., 2014):
$$\frac{dC(t)}{dt} = S(t) - \frac{C(t)}{\tau(t)} \qquad \text{Eq. 5}$$
where $C(t)$ is the atmospheric burden of ammonia at time $t$, $S(t)$ is the time-dependent source
emission fluxes and $\tau(t)$ is the removal timescale. Assuming steady-state conditions and
considering that emission fluxes of ammonia are continuous, there is a quasi-equilibrium
between sources and removals of ammonia (Dentener and Crutzen, 1994), and the modeled
lifetime of ammonia $\tau_{mod}$ can be defined as:
$$\tau_{mod} = C_{NH3}/L_{NH3}^{trans,chem,depo} \qquad \text{Eq. 6}$$
where $C_{NH3}$ is the atmospheric burden of ammonia and $L_{NH3}^{trans,chem,depo}$ is the total loss due to
any process affecting ammonia in the model (transport, chemical reactions, deposition).
We calculate ammonia emission fluxes using IASI satellite measurements that we
interpolated (section 2.2) to the model resolution (2.5°×1.3°) and applying a variable lifetime
taken from a CTM (hereafter NE emissions). We also calculate ammonia emissions from the





oversampled IASI data of Van Damme et al. (2018), after bilinear re-gridding to the model
resolution (2.5°×1.3°), applying a constant lifetime for ammonia of 12 hours (hereafter VD0.5
emissions) and a variable lifetime from a CTM (hereafter VDgrlf emissions).

**2.4    LMDz-OR-INCA chemistry transport model**

The Eulerian global CTM LMDz-OR-INCA was used to calculate ammonia lifetime, as
well as to simulate ammonia concentrations from the emission fluxes calculated from IASI
satellite products. The model couples the LMDz (Laboratoire de Météorologie Dynamique)
General Circulation Model (GCM) (Hourdin et al., 2006) with the INCA (INteraction with
Chemistry and Aerosols) model (Folberth et al., 2006; Hauglustaine et al., 2004) and with the
land surface dynamical vegetation model ORCHIDEE (ORganizing Carbon and Hydrology In
Dynamic Ecosystems) (Krinner et al., 2005). In the present configuration, the model has a
horizontal resolution of 2.5°×1.3°, the vertical dimension is divided into 39 hybrid vertical
levels extending to the stratosphere. Large-scale advection of tracers is calculated from a
monotonic finite-volume second-order scheme (Hourdin and Armengaud, 1999), deep
convection is parameterized according to the scheme of Emanuel, (1991), while turbulent
mixing in the planetary boundary layer (PBL) is based on a local second-order closure
formalism. More information and a detailed evaluation of the GCM can be found in Hourdin et
al. (2006).
The model simulates atmospheric transport of natural and anthropogenic aerosols
recording both the number and the mass of aerosols. The aerosol size distribution is represented
using a modal approach that consists of the superposition of 5 log-normal modes that represent
both the size spectrum and whether the aerosol is soluble or insoluble (Schulz, 2007). The
aerosols are treated in three particle modes, sub-micronic (diameter < 1 μm) corresponding to
the accumulation mode, micronic (diameter 1–10 μm) corresponding to coarse particles, and
super-micronic or super coarse particles (diameter > 10 μm). LMDz-OR-INCA accounts for
emissions, transport (resolved and sub-grid scale), and dry and wet (in-cloud/below-cloud
scavenging) deposition of chemical species and aerosols interactively. LMDz-OR-INCA
includes a full chemical scheme for the ammonia cycle and nitrate particle formation, as well
as a state-of-the-art $CH_4/NO_x/CO/NMHC/O_3$ tropospheric photochemistry. Further details
about specific reactions, reaction rates and other information entering into the description of
the ammonia cycle can be found in Hauglustaine et al. (2014).



The global transport of ammonia was simulated from 2007 to 2017 (2007 was the spin-
up period) by nudging the winds of the 6-hourly ERA Interim Reanalysis data (Dee et al., 2011)
with a relaxation time of 10 days (Hourdin et al., 2006). For the calculation of ammonia's
lifetime, the model ran with traditional emissions for anthropogenic, biomass burning and
oceanic emission sources using emissions from ECLIPSEv5-GFED4-GEIA (hereafter called
EGG) (Bouwman et al., 1997; Giglio et al., 2013; Klimont et al., 2017) .

## 3  Results

In this section, the main results of the monthly emissions (NE) are presented for the 10-
year period (2008–2017) of IASI observations. We first describe the simulated ammonia
lifetimes (section 3.1). Then, we explain the main characteristics of the obtained emissions
(section 3.2) and compare them with those calculated using the IASI gridded products from
Van Damme et al. (2018) (VD0.5 and VDgrlf), as well as the ones from the state-of-the-art
inventories of EGG and EDGARv4.3.1-GFED4 (Crippa et al., 2016; Giglio et al., 2013) that
are often used in CTMs (section 3.3). We finally turn our focus to hot-spot regions and
document their seasonal variation in emissions (section 3.4).

### 3.1  Modelled lifetime of ammonia

The lifetime of ammonia has been reported to range from a few hours to a few days
(Behera et al., 2013; Pinder et al., 2008) so ammonia can only be transported over relatively
short distances. This short spread of ammonia is also due to the fact that (a) the majority of its
emissions are surface ones (major source is agricultural activity), and (b) its surface deposition
velocities are high for most surfaces (Hov et al., 1994). The atmospheric lifetimes of ammonia
were summarized in Van Damme et al. (2018). Specifically, Quinn et al. (1990) and more
recently Norman and Leck (2005) reported lifetimes of a few hours in the West Pacific, South
Atlantic and Indian Oceans, which is in agreement with Flechard and Fowler (1998), who
reported a 2-hour lifetime in an area of Scotland where most sources are of agricultural origin.
Similar to them, Dammers et al. (2019) recently reported a lifetime estimated from satellite
measurements of 2.35±1.16 hours for large point sources based on satellite measurements. The
majority of ammonia lifetimes reported regionally or globally fall within 10 and 24 hours
independently of the different approaches (Hauglustaine et al., 2014; Hertel et al., 2012; Möller
and Schieferdecker, 1985; Sutton et al., 1993; Whitburn et al., 2016b), while Dentener and
Crutzen (1994) reported slightly higher lifetimes within a range between 0.9 and 2.1 days
depending on ammonia emission fraction of natural origin. Monthly averaged atmospheric



ammonia lifetimes in the present study were derived using the version of the LMDz-OR-INCA
that includes non-methane hydrocarbons (Hauglustaine et al., 2004).
Ammonia lifetime depends on numerous factors such as the presence of ammonia's
reactants (sulfuric and nitric acid), meteorological parameters (atmospheric water vapour, and
temperature, atmospheric mixing and advection) and ammonia emissions. In ammonia–poor
conditions, all ammonia is rapidly removed by neutralising sulfuric acid with an intermediate
production of bisulfate. If ammonia increases further (ammonia–rich conditions), then reaction
with nitric acid occurs forming nitric ammonium. At this point, the ammonia/sulfuric acid/nitric
acid equilibrium becomes very fragile. If sulfate concentrations decrease, then free ammonia is
produced, which gradually reacts with nitric acid resulting in production of aerosol phase nitric
ammonium. But if particles are aqueous, then sulfate ions in solution increase the equilibrium
vapour pressure of ammonia with nitric acid reversing the reaction towards gaseous phase
reactants. So, sulfate reductions are linked with non-linear increases of aerosol nitrates and
decreases of aerosol ammonium and water (Seinfeld and Pandis, 2000).
The calculated ammonia lifetime is shown in **Figure 1**a. The average lifetime was
calculated to be 11.6±0.6 hours, which is in the range of the previously reported values. Lower
values (~10 hours) were observed in clean remote areas characterized by low ammonia
emissions (e.g., Amazon forest, Sahara and Australia), while in the rest of the globe the lifetime
was closer to the average value. The highest lifetimes (~16 hours) occur over Southern Brazil
and Venezuela, which are both areas with relatively high ammonia emissions and low sulfuric
and nitric acid concentrations (**Figure 1**c). These conditions are characterized by a low
atmospheric sulfuric and nitric acids availability to remove ammonia rapidly, hence causing an
increase in lifetime.
**3.2   Satellite-constrained emissions**
The average ammonia emissions calculated from the 10-year IASI observations are
shown in **Figure 1**b (also in **Figure S 1**a), the reactants' atmospheric burden in **Figure 1**c and
their seasonal variability in **Figure 1**d. The year-by-year total ammonia emissions are depicted
in **Figure S 1** with a monthly temporal resolution. Emissions decline from 242 Tg yr$^{-1}$ in 2008
to 212 Tg yr$^{-1}$ in 2011. In 2012 – 2014, emissions show little variation (194, 204 and 195 Tg
yr$^{-1}$, respectively), before they increase steeply to 248 Tg yr$^{-1}$ in 2015. Finally, in 2016 and 2017
they remain at the same high level (197 and 227 Tg yr$^{-1}$, respectively).



The global average annual emission calculated from VD0.5 amounts to 189 Tg (9-year
average), which is comparable to the average of the 10-year period that we have calculated in
the present study (average±sd: 213±18.1 Tg yr$^{-1}$). The increase in the emissions we calculate
during 2015 and 2017 stand out. The explanation for these increases could be twofold. If sulfur
dioxide emissions decreased over time, less sulfates are available to neutralize ammonia, hence
resulting in higher ammonia column concentrations seen by IASI that could be attributed to
new emissions erroneously (see section 2.3). If sulfur dioxide and sulfates presented a constant
year-by-year pattern or even increased, then the calculated ammonia emissions would be likely
realistic.
To sort out between these two possibilities, we used sulfur dioxide measurements from
NASA's Ozone Monitoring Instrument (OMI, Yang et al., 2007) instrument, whereas sulfate
column concentrations were taken from the Modern-Era Retrospective Analysis for Research
and Applications, Version 2 (MERRA2, Gelaro et al., 2017) reanalysis data from NASA's
Global Modeling and Assimilation Office (GMAO). **Figure S 2** shows timeseries of column
concentrations of sulfur dioxide and sulfates from OMI and MERRA2 averaged globally, for
continental regions (Europe, North America, South America, Africa), as well as for regions
where ammonia emissions are particularly high (India and Southeastern Asia, North China
Plain). Although column concentrations of both sulfur dioxide and sulfates present strong
interannual variability, they do not show significant changes on an annual basis. This indicates
that sulfate amounts that neutralize ammonia and form ammonium sulfate are rather constant
from year to year and, thus it is likely that the higher ammonia concentrations retrieved from
IASI after 2015 are the result of an emission increase.
Another region of interest is the North China Plain, as it has been identified as an
ammonia hotspot mainly due to extensive agricultural activities (Clarisse et al., 2009; Pan et
al., 2018). Liu et al. (2018) reported a sulfur dioxide reduction of about 60% over the recent
few years in the North China Plain, sulfates decreased by 50%, while ammonia emissions
declined by only 7% due to change in agricultural practices. The suggested decrease in
ammonia reactants over the North China Plain is illustrated by the calculated sulfur dioxide
column concentration anomaly from OMI (**Figure 2**) and by the sulfate concentration anomaly
from MERRA-2 after 2015 (the highest calculated one) (**Figure S 3**). However, the IASI-
constrained ammonia emissions calculated here show only a tiny increase of 0.19±0.04 kt y$^{-1}$
after 2015 in the North China Plain and of 10±3.1 Tg y$^{-1}$ globally with respect to the 10-year





average (Figure 2). This is due to the change of SO2 and NOx emission regulations in China,
which in turn led to reduced inorganic matter (sulfates, nitrates and ammonium) resulting in
regional increases of gaseous ammonia (Lachatre et al., 2019).

### 3.3 Comparison with traditional emission datasets

In this section, we quantify the main differences of our IASI-constrained emission dataset
with other state-of-the-art inventories used in global models and for different applications (air
quality, climate change etc…). Aside from comparing our emissions with those calculated using
Van Damme et al. (2018) data with a constant lifetime (hereafter called VD0.5), we extend our
comparison to more traditional datasets such as those of ECLIPSEv5-GFED4-GEIA (EGG) for
2008–2017, and EDGARv4.3.1-GFED4 (Crippa et al., 2016; Giglio et al., 2013) for 2008–2012
period. Finally, the ammonia emissions presented in this study (NE emissions) are compared
to emissions calculated from Van Damme et al. (2018) gridded IASI column data applying a
variable (modelled) ammonia lifetime presented in Figure 1b (hereafter referred as VDgrlf).
The 10-year comparison of our calculated emissions with VD0.5 is shown in Figure 3.
The 10-year average difference amounts to $29\pm15$ Tg yr$^{-1}$ (average±sd). In all years, the largest
differences could be seen over Latin America and over tropical Africa. Our emissions (NE)
were lower in the Indo-Gangetic Plain, and situated slightly more northerly. Northern India has
been previously identified as a hot-spot region for ammonia, mainly due the importance of
agricultural activities in the region (Kuttippurath et al., 2020; Tanvir et al., 2019).
Figure S 4 and Figure S 5 present a comparison of our calculated emissions with the basic
state-of-the-art datasets of EGG and EDGARv4.3.1-GFED4, respectively. In both datasets,
ammonia emissions remain almost constant over time (average±sd: $65\pm2.8$ Tg yr$^{-1}$ and $103\pm5.5$
Tg yr$^{-1}$, respectively). The total calculated ammonia emissions are up to three times lower than
those calculated from NE (average±sd: $213\pm18.1$ Tg yr$^{-1}$) or from VD0.5 (9-year average: 189
Tg yr$^{-1}$). This results in 10-year annual differences that are very significant (average±sd:
$150\pm19.3$ Tg yr$^{-1}$ and $111\pm19.2$ Tg yr$^{-1}$, respectively); the largest differences appear over South
America, while European emissions are practically identical in all datasets. Emissions from
South China Plain are much higher in the two traditional datasets that those calculated here.
Based upon IASI retrievals, Liu et al. (2019) showed an increase of surface NH$_3$ concentrations
trend of more than 0.2 µg N m$^{-3}$ yr$^{-1}$) in eastern China and of around 0.1–0.2 µg N m$^{-3}$ yr$^{-1}$ in
northern Xinjiang during 2008–2016. Ammonia emissions derived over China in this work are



among the highest worldwide (**Figure S 1**), which agrees well with the 9-year average emissions
calculated in VD0.5 inventory over China (see **Figure 3**). To assess to which extent emissions
from EGG and EDGARv4.3.1-GFED4 are underestimated can only be done by comparing
ammonia with ground or satellite observations.

370          The comparison of the annual ammonia emissions to the modified VDgrlf emissions is

shown in **Figure S 6**. The latter showed a better agreement to the emissions presented in this
study with mean annual different of $14\pm19$ Tg yr$^{-1}$ (average±sd). Previously observed emission
differences in the two state-of-the-art inventories over South America and Africa have been
now minimized, as well as the displacement north of the Indo-Gangetic Plain emissions remains
important. Nevertheless, the smaller differences of our emissions from those of VDgrlf as
compared with the respective difference from the VD0.5 emissions, show the large impact that
a more realistic variable lifetime might have in emission calculations with this methodology.

### 3.4   Site-specific ammonia emissions and seasonal variation

379          **Figure 4** illustrates specific regions that show the largest ammonia emissions (Europe,

North America, South America and Southeastern Asia). These emissions correspond to the
IASI-constrained emissions calculated in this study (NE) and are presented as total annual
emissions averaged over the 10-year period of study. At the bottom panels of the same figure,
the seasonal variation of the emissions is shown for each of the four hot-spot regions and each
of the 10 years of the study.

385          European total ammonia emissions were estimated to be $15\pm2.2$ Tg yr$^{-1}$ (average±sd),

more than double compared with those reported in EGG ($6.9\pm1.1$ Tg yr$^{-1}$) and similar to those
in VD0.5 (11 Tg yr$^{-1}$) or those in VDgrlf ($11\pm1.0$ Tg yr$^{-1}$). The greatest emissions were
calculated for Belgium, the Netherlands and the Po Valley in Italy (**Figure 4**). High emissions
are also found in North and Northwestern Germany and over Denmark. In contrast, very low
emissions are found in Norway, Sweden and parts of the Alps. It is not possible to quantitatively
distinguish between different sources of ammonia. It has been reported that approximately 75%
of ammonia emissions in Europe originate from livestock production (Webb et al., 2005), and
90% from agriculture in general (Leip et al., 2015). More specifically, ammonia is emitted from
all stages of manure management, from livestock buildings during manure storage and
application to land, as well as from livestock urine. These emissions are strong over most of
Northwestern European countries, although sources like fertilization and non-agricultural



397 activities (traffic and urban emissions) can be also important. An example is Tange in Germany,

398 which shows a late summer peak due to growing crops application. No obvious seasonality in

399 the emissions can be seen for Europe as a whole, as the hot-spot regions are rather few compared

400 to the overall surface of Europe. An exception to this stable emission situation over the year

401 occurs during 2010 and during 2015, years for which a late summer peak. In 2010, large

402 wildfires in Russia resulted in high ammonia emissions (R'Honi et al., 2013), while year 2015

403 has been also characterized as an intense fire year (though not like 2010), with fires occurring

404 in Eurasia (Min Hao et al., 2016).

405 North America and in particular the US (**Figure 4**) has been characterized by four hot-

406 spot regions. First, a small region in Colorado, Central US, which is the location of a large

407 agricultural region that traditionally releases large ammonia emissions (Malm et al., 2013).

408 Another example is the state of Iowa (home to more than 20 million swine, 54 million chickens,

409 and 4 million cattle), northern Texas and Kansas (beef cattle), and southern Idaho (dairy cattle)

410 (McQuilling, 2016). Furthermore, the three major valleys in Salt Lake, in Cache, and in Utah

411 in the midwestern US show an evident, but lower intensity hot-spot, as they are occupied by

412 massive pig farms associated to open waste pits. The largest emissions were calculated for the

413 San Joaquin Valley in California (vegetables, dairy, beef cattle and chickens) and further to the

414 South (Tulare and Bakersfield), an area characterized by feedlots (Van Damme et al., 2018;

415 McQuilling, 2016). North American annual ammonia emissions over the 10-year period were

416 averaged $1.1\pm0.1$ Tg yr$^{-1}$ (average±sd). These values are over two orders of magnitude higher

417 than those in EGG ($0.062\pm0.0013$ Tg yr$^{-1}$). Note that his estimate is three times lower than those

418 reported in VD0.5 (3.1 Tg yr$^{-1}$) or in VDgrlf ($3.4\pm0.5$ Tg yr$^{-1}$). The 2008–2017 interannual

419 variability (**Figure 4**) all show a minimum in winter. Maximum emissions were observed in late

420 spring, due to the contribution from mineral fertilizer and manure application, in summer, due

421 to influence of livestock housing emissions, and some years both in spring and summer (Makar

422 et al., 2009; Zhu et al., 2013, 2015). A topographical dependence was also seen in midwest

423 emissions that peaked in April, whereas over the rest of the US maximum emissions were

424 appeared in summer (Paulot et al., 2014).

425 Ammonia emissions have different characteristics in South America and in Western

426 Africa as both are fire-dominated regions. For simplicity we only present South America in

427 **Figure 4**. This region is dominated by natural ammonia emissions mainly from forest, savanna

428 and agricultural fires (Whitburn et al., 2014, 2016b) and volcanoes (Kajino et al., 2004;



Uematsu et al., 2004). This causes a strong seasonal variability in the ammonia emissions with
the largest fluxes observed from August to October in all years (**Figure 4**). This strong
dependence of South America from biomass burning emissions was first highlighted by Chen
et al. (2013) and by van Marle et al. (2017). It also became particularly pronounced during the
large wildfires in the Amazon rainforest in summer 2019 (Escobar, 2019). We estimated the
10-year average ammonia emissions to be $28\pm3.0$ Tg yr$^{-1}$ (average±sd) in agreement with
VD0.5 (22 Tg yr$^{-1}$) and VDgrlf ($24\pm1.3$ Tg yr$^{-1}$). The respective emissions in EGG are four
times lower than these estimates ($7.1\pm0.3$ Tg yr$^{-1}$).
The last column to the right of **Figure 4** presents the 10-year average annual ammonia
emissions and their respective interannual variability in Southeastern Asia. We define this
region spanning from 70°E–130°E in longitude and from 0°N–45°N in latitude. Ammonia
emissions were estimated to be $38\pm2.8$ Tg yr$^{-1}$ (average±sd) similar to VD0.5 (36 Tg yr$^{-1}$) and
VDgrlf ($39\pm1.8$ Tg yr$^{-1}$) and slightly higher than those presented in EGG ($25\pm1.2$ Tg yr$^{-1}$). They
comprise ammonia fertilizer plants, such as in Pingsongxiang, Shizuishan, Zezhou-Gaoping,
Chaerhan Salt Lake, Delingha, Midong-Fukang and Wucaiwan (China), Indo-Gangetic Plain
(Pakistan and India), Gresik (Indonesia). China and India contribute more than half of total
global ammonia emissions since the 1980s with the majority of these emissions to originate
from rice cultivation followed by corn and wheat (crop-specific emissions). More specifically,
emissions from these crops due to synthetic fertilizer and livestock manure applications are
concentrated in North China Plain (Xu et al., 2018). Considering that Southeastern Asia is the
largest agricultural contributor in the global ammonia budget, a strong seasonality in the
emissions was observed. Temporal ammonia emissions peak in late summer of most years,
when emissions from rice cultivation, synthetic fertilizer application and livestock manure
spreading (Xu et al., 2016) are important, and in spring when wheat cultivation dominates
(Datta et al., 2012). Of course, the respective emissions from biomass burning should also be
mentioned. However, these are difficult to be distinguish and are expected to be a relatively
small source compared to agricultural emissions.

## 456    4  Discussion

In this section, we conduct simulations over the 10-year period (2008–2017, 1-year spin-
up), with all the emissions derived and compare the NH$_3$ concentrations with ground based
observations over Europe, North America, Southeastern Asia (section 4.1), and observations
from CrIS (section 4.2). These simulations consist of: (i) a simulation using traditional



emissions from ECLIPSEv5 (Evaluating the CLimate and Air Quality ImPacts of Short-livEd
Pollutants) for anthropogenic sources, GFED4 (Global Fire Emission Dataset) for biomass
burning emissions and GEIA (Global Emissions InitiAtive) for oceanic sources from
(Bouwman et al., 1997; Giglio et al., 2013; Klimont et al., 2017), where modelled lifetimes of
ammonia were also calculated (EGG); (ii) a simulation using emissions calculated from IASI
data from Van Damme et al. (2018) applying a constant lifetime of 12 hours for ammonia
(VD0.5); (iii) a simulation using gridded emissions presented in the present paper (NE)
calculated as described in section 2; and (iv) a simulation using emissions calculated from IASI
data from Van Damme et al. (2018) applying a variable (modelled) lifetime (VDgrlf). Finally,
we perform a sensitivity analysis in order to define the levels of uncertainty of our emissions in
section 4.3 and discuss potential limitation of the present study in section 4.4.

### 4.1   Validation against ground-based observations

**Figure 5** shows a comparison between modelled surface concentrations of ammonia with
ground measurements from Europe (EMEP, https://emep.int/mscw/), North America (AMoN,
http://nadp.slh.wisc.edu/data/AMoN/)          and          Southeastern          Asia          (EANET,
https://www.eanet.asia). To avoid overplotting, the Gaussian kernel density estimation (KDE)
was used, which is a non-parametric way to estimate the probability density function (PDF) of
a random variable (Parzen, 1962):
$$f(x) = \frac{1}{Nh}\sum_{i=1}^{N}K(\frac{x-x_i}{h})$$          **Eq. 7**
where $K$ is the kernel, $x_i$ the univariate independent and identically distributed point of the
relationship between modelled and measured ammonia and $h$ is a smoothing parameter called
the bandwidth. KDE is a fundamental data smoothing tool that attempts to infer characteristics
of a population, based on a finite dataset. It weighs the distance of all points in each specific
location along the distribution. If there are more points grouped locally, the estimation is higher
as the probability of seeing a point at that location increases. The kernel function is the specific
mechanism used to weigh the points across the data set and it uses the bandwidth to limit the
scope of the function. The latter is computed using the Scott's factor (Scott, 2015). We also
provide the mean fractional bias (MFB) for modelled and measured concentrations of ammonia
as follows:
$$MFB = \frac{1}{N}\frac{\sum_{i=1}^{N}(C_m-C_o)}{\sum_{i=1}^{N}(\frac{C_m+C_o}{2})} \times 100\%$$          **Eq. 8**
where $C_m$ and $C_o$ are the modelled and measured ammonia concentrations and $N$ is the total
number of observations. MFB is a symmetric performance indicator that gives equal weights





to under- or over-estimated concentrations (minimum to maximum values range from -200%
to 200%). Furthermore, we assess the deviation of the data from the line of best fit using the
root mean square error (RMSE) defined as:
$$RMSE = \sqrt{\sum_{i=1}^{N} \frac{(C_m - C_o)^2}{N}}$$    **Eq. 9**
From 134 European stations, nearly 300,000 measurements made at a daily to weekly
temporal resolution over the period of study (2007–2018) are presented on **Figure 5**. All
emission datasets underestimate ammonia surface concentration over Europe. The most
accurate prediction of concentrations was achieved using the traditional EGG emissions that
underestimated observations by 67%, also being the least scattered from the best fit
($RMSE_{EGG} = 4.06\ \mu g\ N\ m^{-3}$), followed by the emissions presented in this paper ($MFB_{NE} =$
$-72\%,\ RMSE_{NE} = 4.65\ \mu g\ N\ m^{-3}$), although they were more variable. VD0.5 or VDgrlf
emissions further underestimated observations, though they were less sparse (**Figure 5**d). About
12% of the modelled concentrations using EGG were outside of the 10-fold limit from the
observations, in contrast to only 17% and 15% in VD0.5 and VDgrlt, and 20% in NE. With
regards to the spatial comparison with the observed concentrations, all datasets cause
overestimations in the ammonia concentrations predicted in Eastern Europe (station
AM0001R). EGG appears to be the most accurate in Central Europe (all stations with suffix
DE00), NE emissions in all Spanish stations (suffix ES00) and VD0.5 and VDgrlf emissions in
Italian stations (**Figure S 7**).
The comparison of simulated ammonia concentrations to observations over North
America includes 119 stations, which represent nearly 27,000 observations (Figure 6) with a
weekly, bi-weekly or monthly resolution. The only emission dataset that lead to an
underestimation of ammonia concentrations was EGG (**$MFB_{EGG} = -28\%$**). Two others,
VD0.5 and VDgrlf caused ammonia observations to be strongly overestimated (**$MFB_{VD0.5} =$**
**$52\%\ and\ MFB_{VDgrlf} = 54$**%), while NE slightly (**$MFB_{NE} = 32\%$**). All inventories resulted
in about the same variability in ammonia concentrations with RMSEs between 4.15 and 4.17
µg N m-3 (Figure 6). About 10% of the predicted concentrations using EGG emissions were at
least 10 times off from the measured ones, more than twice the number of measurements
compared to the other dataset. NE emissions better capture levels in the easternmost stations of
the US (AL99, AR15, CT15, IL37, IN22, MI52, NY56, ON26) and in California (CA83) and
Oklahoma (OK98), which are close to hot-spot regions (see section 3.4). EGG emissions
perform better in Northwestern (ID03), Central (KS03) and several stations located over the



Eastern United States (KY03, KY98, OH09, AR03, IL46, KS03, GA41). The emission
inventory VD0.5 leads to a very good agreement in ammonia concentrations over all stations
of the North American continent (AL99, GA40, ID03, GA41, IL37, IL46, IN20, IN22, KS97,
PA00, MD99, MI52, TN04, NM99, NY96, OH99, OK98) (Figure S 8).

529        In Southeastern Asia 62 stations from 13 countries were included in the comparison from

the EANET monitoring network (Figure 7). These included about 8,000 surface measurements
in monthly or 2-weekly resolution. All emission inventories underestimate station
concentrations of EANET with MFBs beween -102% (EGG) and -61% (VD0.5 and VDgrlf) ().
The least spread model concentrations were those simulated using VD0.5 and VDgrlf
($RMSE = 4.61 - 4.65\ \mu g\ N\ m^{-3}$). Around 19% of model concentrations using EGG were
outside the 10-fold limit of the 1×1 line with observations, 12% using NE emissions and only
5% and 6% using VD0.5 and VDgrlf, respectively. VD0.5 and VDgrlf emissions capture well
the Japanese (suffix JPA) and Taiwanese stations (suffix THA). Given the short lifetime and
the relatively coarse spatial scales, the model fails to capture the variability that exists within
each gridbox (Figure S 9).
**4.2  Validation against satellite products**

541        Here, we used surface ammonia concentrations from CrIS from 1st May 2012 to 31st

December 2017 and we compared them with modelled ammonia concentrations using four
emissions datasets (EGG, VD0.5, NE and VDgrlf), like in the previous section. The comparison
is shown as PDF of surface modelled against CrIS concentrations of ammonia calculated with
the Gaussian KDE in Figure 8. A total of 4.5 million surface measurements were used in the
comparison with a global coverage. All datasets underestimated surface concentrations except
NE emissions, which overestimate ammonia ($MFB = +0.48$). The best fit was achieved for
the VDgrlf emissions, which slightly underestimate ammonia ($MFB = -0.37$), while 82% of
the measurements were within one order of magnitude from the 1×1 line, which is also shown
by the small $RMSE$. VD0.5 emissions produced similar concentrations, with respect to the
$RMSE$ and $MFB$ values, whereas 79% of them were less than a 10-fold difference from the
observations. NE emissions result in higher surface concentrations, also showing larger
$RMSEs$. However, 90% of the modelled concentrations were within a factor of 10 from the
CrIS observation. In general, a better agreement for the most recent years 2015 – 2017 was
achieved. The baseline EGG emissions resulted in significantly larger deviations of modelled



surface concentrations of ammonia from the CrIS observations, as shown in **Figure 8** comprising the largest *RMSE* and *MFB* values.

## 4.3 Uncertainty analysis

A sensitivity analysis in order to calculate the level of uncertainty that each of the parameter gives to the modelled surface concentrations of ammonia was also performed. The relative uncertainty was calculated as the standard deviation of ammonia's surface concentrations from a model ensemble of 10 members (**Table 1**) divided by the average. The first six members are the surface concentrations that resulted from simulations of ammonia emissions after perturbation of the Euclidian distance $d_k$ in the parameters of the IDW interpolation. The remaining four members are simulated concentrations using the previously reported emissions datasets (EGG, VD0.5, NE and VDgrlf). The results are shown as a 10-year (2008–2017) annual average relative uncertainty in **Figure 9** and as annual average relative uncertainty of surface concentrations for every year of the 10-year period in **Figure S 10**.

The surface concentrations resulting from the different calculated emissions mainly affects oceanic regions, with values reaching 100%. The reason for this could be threefold. First, the IDW interpolation shows to be affected by severe outlier values, which are found in several oceanic regions (**Figure S 11**); this creates high gridded column ammonia concentrations and, in turn, fluxes at regions that are not supported by previous findings or measurements. Second, the methodology with which ammonia concentrations are retrieved in IASI has certain limitation, with respect to (i) the use of constant vertical profiles for ammonia, (ii) potential dependencies of total column ammonia and temperature that are not taken into account, and (iii) instrumental noise that can cause a high bias of the measurements (Whitburn et al., 2016a). Third, there is much less ammonia over the Ocean, hence the relative error bars are much larger. Large uncertainties in surface ammonia concentrations were observed in regions characterized by large anthropogenic contribution, such as North India, North China Plain and Central USA. Smaller uncertainties were found in Central Africa and in Amazonia, regions that are linked with episodic biomass burning emissions (**Figure 4**).

## 4.4 Limitations of the present study

We discuss the importance of certain limitations in the methodology of the present study and in the validation of the results. These limitations will also be commented upon in the overall conclusion of the paper.



Regarding the methodology, emissions of short-lived species are determined, among
other methods, using top-down approaches. When only satellite measurements are available,
they are usually averaged over a particular location and surface emissions are calculated using
a mass balance approach (Lin et al., 2010; Zhao and Wang, 2009). This is done by assuming a
1-dimensional box-model, where atmospheric transport between grids is assumed to be
negligible and loss due to deposition or chemical reactions very fast. The solution to this
problem is the use of Kernels (Boersma et al., 2008), which makes the computation of the
emissions very intense. It has been reported that for resolutions, such as those used in the
present paper (2.5°×1.3°), non-local contributions to the ammonia emissions are relatively
small (Turner et al., 2012). Although, the use of Kernels is the proper way to account for non-
local contributions, we believe that negligible transport here is a fair assumption, due to the
small lifetimes of ammonia calculated from the CTM (11.6±0.6 hours); therefore,
transportation from the adjacent grid-cells should be small. Note that already this method has
been suggested for short lived climate pollutants, it is not suitable for species with lifetime from
days to weeks (e.g. black carbon, Bond et al., 2013).
Another limitation of the present study is that the same model is used for the calculation
of the modelled lifetimes and for the validation of the emissions that were calculated using
these lifetimes (NE and VDgrlf). A more accurate validation would require an independent
model for the simulations of surface concentrations using these emissions.

## 5  Conclusions

In the present paper, satellite measurements from IASI were used to constrain global
ammonia emissions over the period 2008–2017. The data were firstly processed to monthly
ammonia column concentrations with a spatial resolution of 2.5°×1.3°. Then, using gridded
lifetime for ammonia calculated with a CTM, monthly fluxes were derived. This contrasts with
previously reported methods that used a single constant lifetime. This enables a more accurate
calculation in regions sensitive to the changing balance between nitrate and sulfate abundances.
The calculated ammonia emission fluxes were then used to simulate ammonia concentrations
for the period 2008–2017 (referred to as NE). The same simulations were repeated using
baseline emissions from ECLIPSEv5-GFED4-GEIA (referred to as EGG), emissions
constrained by Van Damme et al. (2018) IASI data using a constant lifetime for ammonia
(named as VD0.5) and emissions based on Van Damme et al. (2018) retrievals using a modelled
lifetime from a CTM (named as VDgrlf). The simulated surface concentrations of ammonia
were compared with ground measurements over Europe (EMEP), North America (AMoN) and





Southeastern Asia (EANET), as well as with global satellite measurements from CrIS. The
main conclusions can be summarized as follows:
• The 10-year average annual ammonia emissions calculated here (NE) were estimated to be
$213\pm18.1$ Tg yr$^{-1}$, which is 15% higher than those in VD0.5 (189 Tg yr$^{-1}$), and 6% higher
than those in VDglrf ($201\pm10.4$ Tg yr$^{-1}$). These emission values amount to twice the
published from datasets, such as EGG ($65\pm2.8$ Tg yr$^{-1}$) and EDGARv4.3.1-GFED4,
($103\pm5.5$ Tg yr$^{-1}$).
• In the North China Plain, a region characterized by intensive agricultural activities, a small
increase of ammonia emissions is simulated after 2015. This is attributed to decreases in
sulfur species, as revealed from OMI and MERRA-2 measurements. Less sulfates in the
atmosphere leads to less ammonia neutralization and hence to larger loads in the
atmospheric column as measured by IASI.
• In Europe, the 10-year average of ammonia emissions were estimated at $15\pm2.2$ Tg yr$^{-1}$
(NE), twice as much as those in EGG ($6.9\pm1.1$ Tg yr$^{-1}$) and similar to those in VD0.5 (11
Tg yr$^{-1}$) or VDgrlf ($11\pm1.0$ Tg yr$^{-1}$). The strongest emission fluxes were calculated over
Belgium, Netherlands, Italy (Po Valley), Northwestern Germany and Denmark. These
regions are known for industrial and agricultural applications, animal breeding activities,
manure/slurry storage facilities and manure/slurry application to soils.
• Some hot-spot regions with high ammonia emissions were distinguished in North America:
(i) in Colorado, due to large agricultural activity, (ii) in Iowa, northern Texas and Kansas,
due to animal breeding, (iii) in Salt Lake, Cache, and Utah, due to animal farms associated
with open waste pits and (iv) in California, due to animal breeding and agricultural
practices. Ammonia emissions in North America were $1.1\pm0.1$ Tg yr$^{-1}$ or two orders of
magnitude higher than in EGG ($6.2\pm0.1$ kt yr$^{-1}$) and three times lower than those in VD0.5
($3.1$ Tg yr$^{-1}$) or in VDgrlf ($3.4\pm0.5$ Tg yr$^{-1}$), with maxima observed in late spring, due to
fertilization and manure application and summer, due to livestock emissions.
• South America is dominated by natural ammonia emissions mainly from forest, savanna
and agricultural fires causing a strong seasonality with the largest fluxes between August
and October. The 10-year average ammonia emissions were as high as $28\pm3.0$ Tg yr$^{-1}$
similar to VD0.5 (22 Tg yr$^{-1}$) and VDgrlf ($24\pm1.3$ Tg yr$^{-1}$) and four times higher than EGG
($7.1\pm0.3$ Tg yr$^{-1}$).
• In Southeastern Asia, the 10-year average ammonia emissions were $38\pm2.8$ Tg yr$^{-1}$, in
agreement with VD0.5 (36 Tg yr$^{-1}$) and VDgrlf ($39\pm1.8$ Tg yr$^{-1}$) and slightly higher than





those in EGG (25±1.2 Tg yr⁻¹). The main sources were from fertilizer plants in China,
Pakistan, India and Indonesia. China and India hold the largest share in the ammonia
emissions mainly due to rice, corn and wheat cultivation. A strong seasonality in the
emissions was observed with a late summer peak in most years, due to rice cultivation,
synthetic fertilizer and livestock manure applications and in spring due to wheat
cultivation.
• About 88% of the modelled concentrations over Europe using EGG were inside the 10-
fold limit from the observations, higher than those with VD0.5 (83%), VDgrlf (85%) and
NE (80%). All emission datasets overestimate of ammonia in Eastern Europe, EGG
captures better Central Europe, NE emissions predict concentrations in Spain and VD0.5
with VDgrlf emissions in Italy.
• In North America, 90% of the modelled concentrations using EGG emissions were less
than 10 times different from the measured ones; more than 95% of the modelled
concentrations in North American stations were in the same range using NE, VD0.5 and
VDgrlf emissions. NE emissions better capture levels in the easternmost stations of the US
closer to the respective hot-spot regions, whereas EGG emissions perform better in
Northwestern and Central USA. VD0.5 and VDgrlf emissions perform well in most of the
North American stations.
• All emissions underestimate station concentrations in Southeastern Asia. The least spread
model concentrations were those simulated using VD0.5 and VDgrlf. About 81% of
modelled concentrations using EGG were in the 10-fold limit of the 1×1 line with
observations, 88% using NE and only 95% and 94% using VD0.5 and VDgrlf, respectively.
VD0.5 and VDgrlf emissions capture well the Japanese and Taiwanese stations.
• The comparison of the modelled ammonia with satellite observations from CrIS globally
showed that the best agreement was achieved using the VDgrlf emissions in 2012–2014.
After 2015, all satellite retrieved emissions show a better agreement with CrIS
concentrations.
Overall, the satellite-constrained ammonia emissions calculated using a variable lifetime
appear to give more realistic concentrations, with respect to station and satellite measurements.
Accordingly, state-of-the-art emissions appear to underestimate ammonia significantly.



*Data availability.* All data and python scripts used for the present publication are open through the web address https://folk.nilu.no/~nikolaos/AMMONIA/ or can be obtained from the corresponding author upon request.

*Competing interests.* The authors declare no competing interests.

*Acknowledgements.* This study was supported by the Research Council of Norway (project ID: 275407, COMBAT – Quantification of Global Ammonia Sources constrained by a Bayesian Inversion Technique). Lieven Clarisse and Martin Van Damme are respectively a research associate and a postdoctoral researcher supported by the F.R.S.–FNRS.

*Author contributions*. N.E. performed the simulations, analyses, wrote and coordinated the paper. S.E. contributed to the lifetime calculations. Y.B., D.H. and A.C. set up the CTM model. M.V.D., P.-F.C. and L.C. provided the IASI data, while M.W.S. and K.E.C.-P. provided the observations from CrIS. All authors contributed to the final version of the manuscript.

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





**FIGURE LEGENDS**

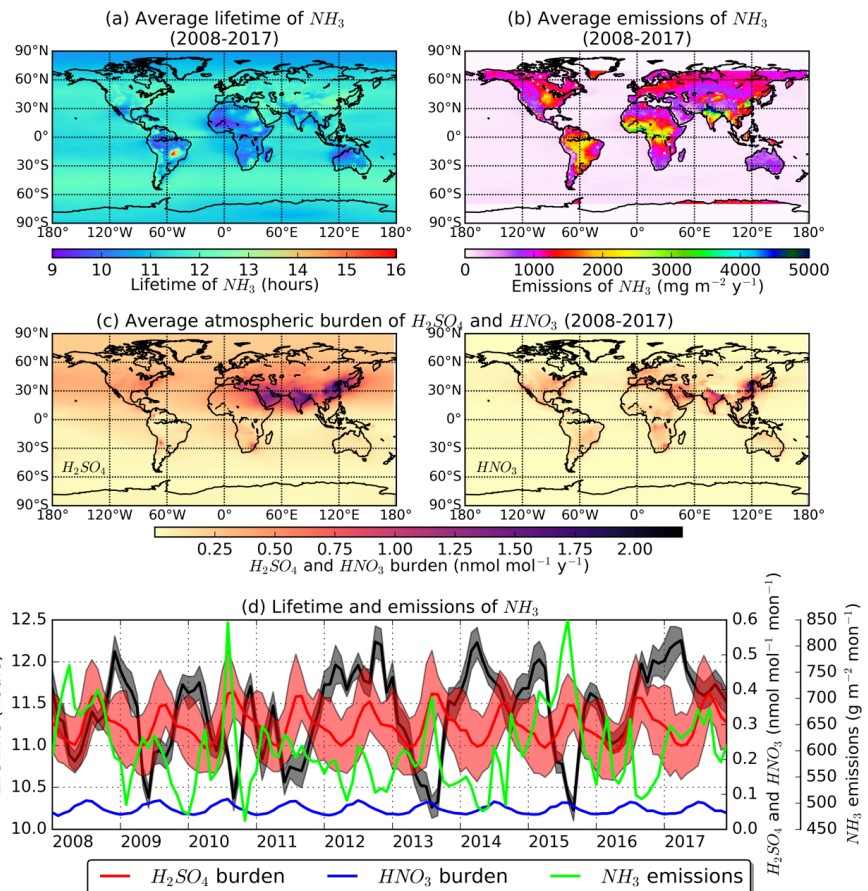

Figure 1. (a) Average model lifetime of ammonia calculated from the LMDz-OR-INCA model,
(b) total annual emissions averaged over the 10-year period, (c) atmospheric burden of the
reactants sulfuric and nitric acid calculated in the model, and (d) monthly timeseries of lifetime
(black), ammonia emissions (green), sulfate (red) and nitrate column concentrations (blue) for
the whole 10-year period.

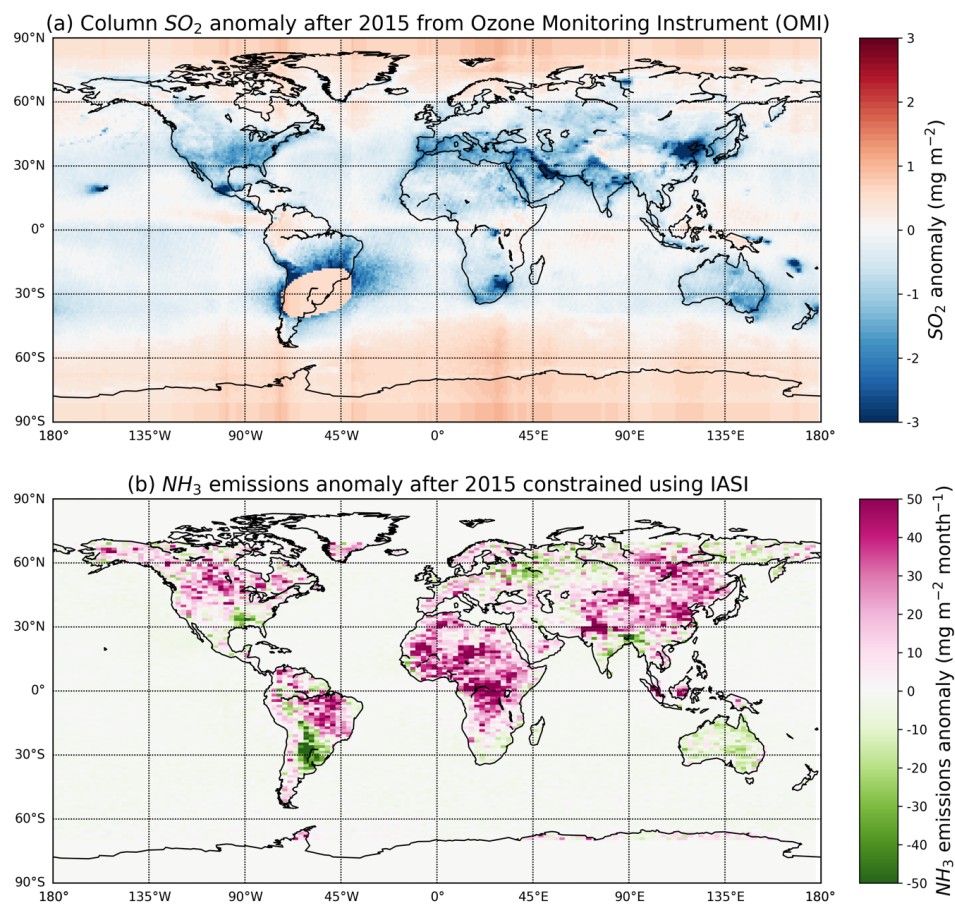


Figure 2. (a) Annual average total column sulfur dioxide anomaly after 2015 from OMI, and
(b) annual average emission anomaly of ammonia calculated from IASI in the present study
(NE).

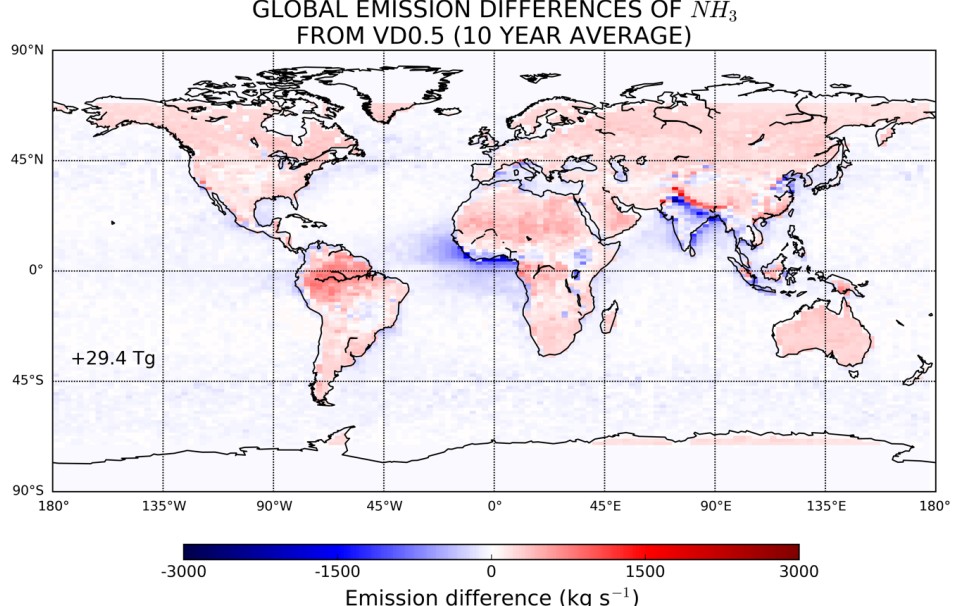

Figure 3. Global differences of ammonia emissions calculated in the present study (NE) from those calculated using Van Damme et al. (2018) gridded concentrations applying a constant lifetime of 0.5 days (VD0.5). The results are given as 10-year average (2008–2017) and the number denotes the annual difference in the emissions.



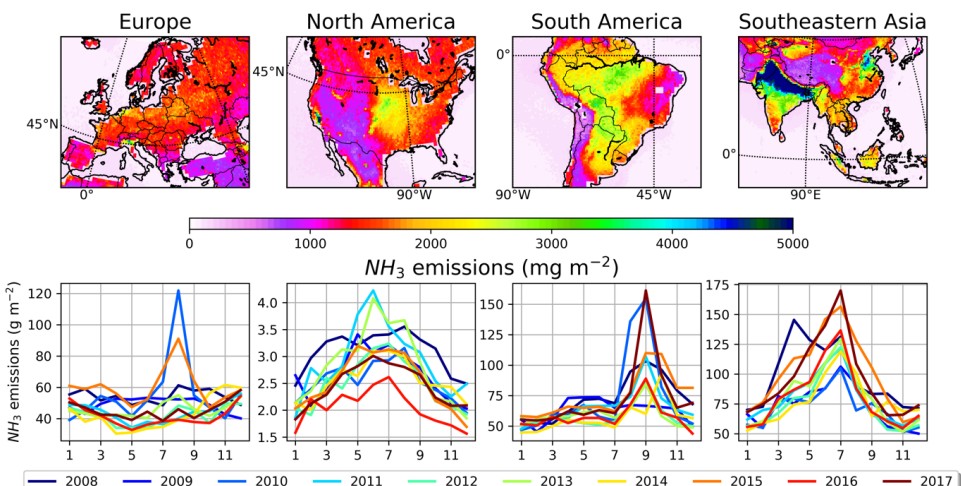

Figure 4. Total annual emissions of ammonia averaged over the 10-year period (2008–2017) in
Europe, North and South America and Southeastern Asia, which are regions characterized by
the largest contribution to global ammonia budget. In the bottom panels the monthly variation
of the emissions is shown for each year of the study period.

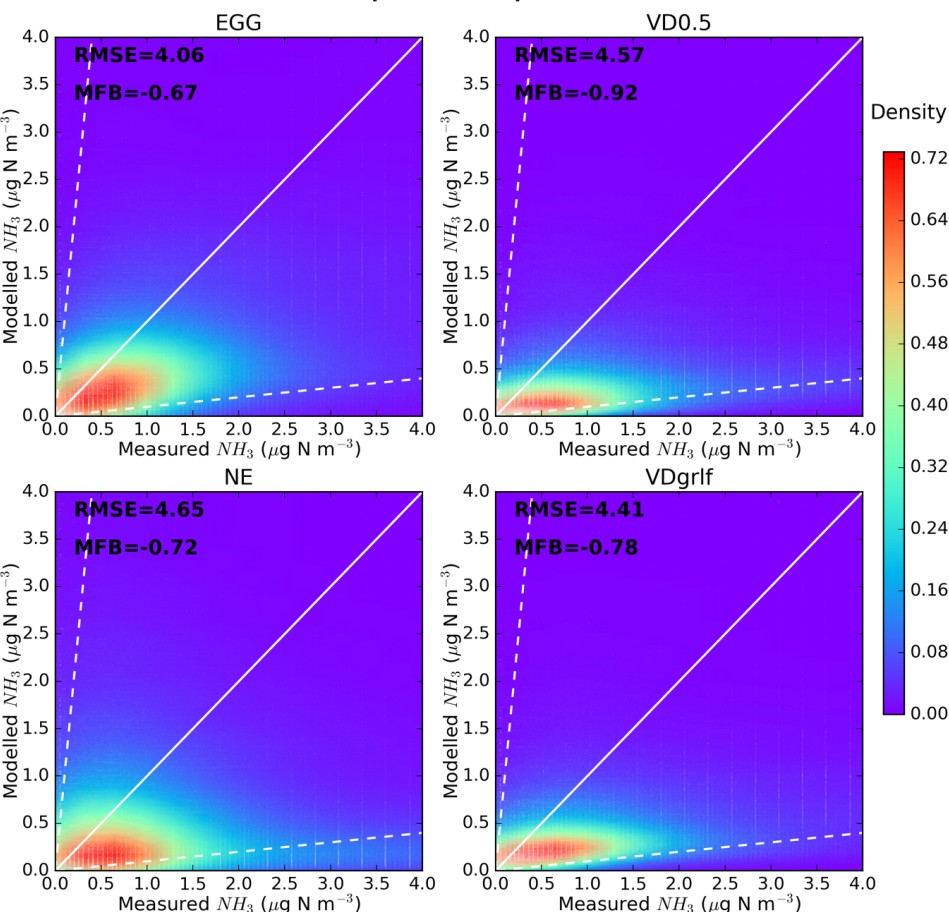

Figure 5. Validation of modelled concentrations of ammonia for different emissions datasets
(EGG, VD0.5, NE and VDgrlf) against ground-based measurements from EMEP for the 10-
year (2008–2017) study period. Scatterplots of modelled against measured concentrations for
the aforementioned emission inventories were plotted with the Kernel density estimation, which
is a way to estimate the probability density function (PDF) of a random variable in a non-
parametric way.



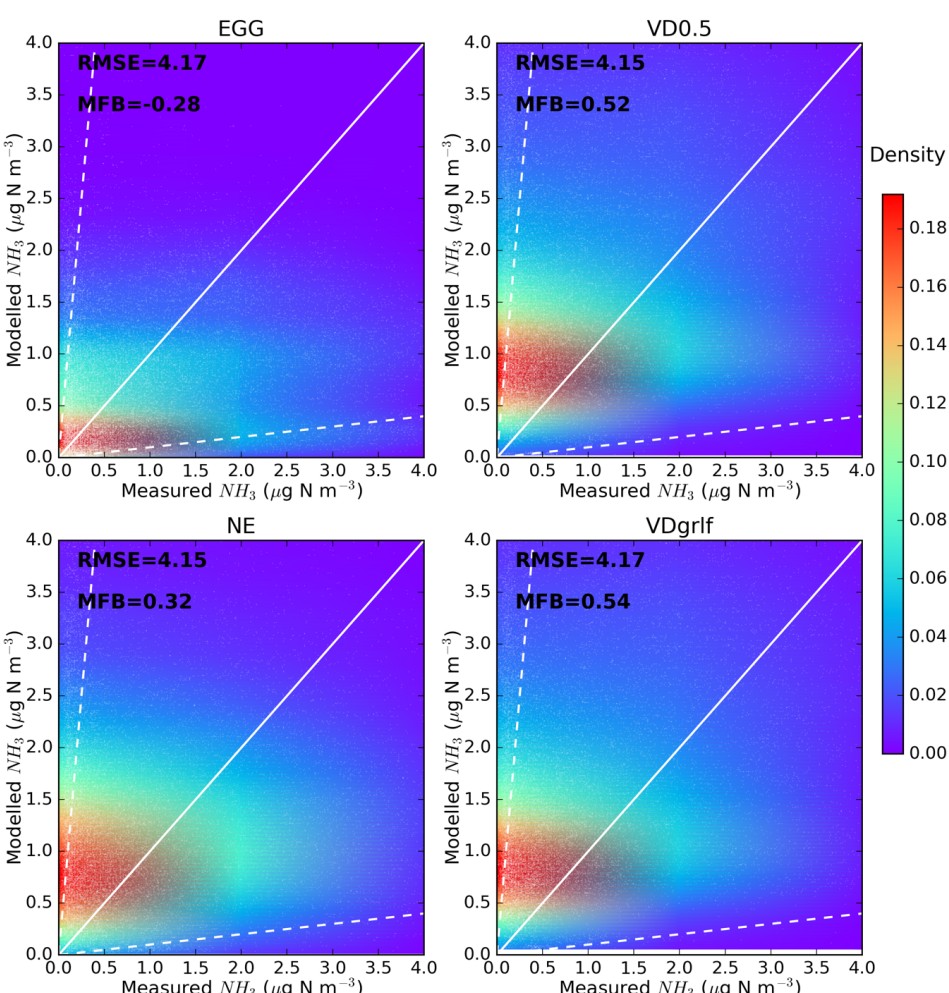


Figure 6. Validation of modelled concentrations of ammonia for different emissions datasets (EGG, VD0.5, NE and VDgrlf) against ground-based measurements from AMON for the 10-year (2008–2017) study period. Scatterplots of modelled against measured concentrations for the aforementioned emission inventories were plotted with the Kernel density estimation, which is a way to estimate the probability density function (PDF) of a random variable in a non-parametric way.




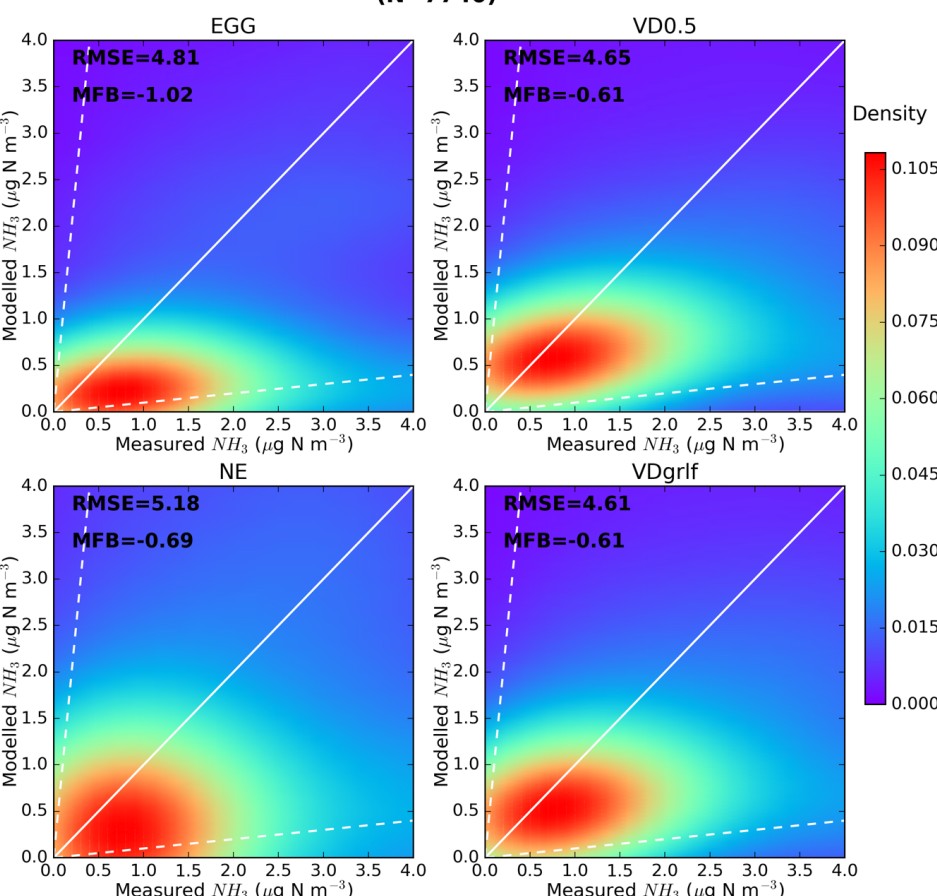


Figure 7. Validation of modelled concentrations of ammonia for different emissions datasets (EGG, VD0.5, NE and VDgrlf) against ground-based measurements from EANET for the 10-year (2008–2017) study period. Scatterplots of modelled against measured concentrations for the aforementioned emission inventories were plotted with the Kernel density estimation, which is a way to estimate the probability density function (PDF) of a random variable in a non-parametric way.





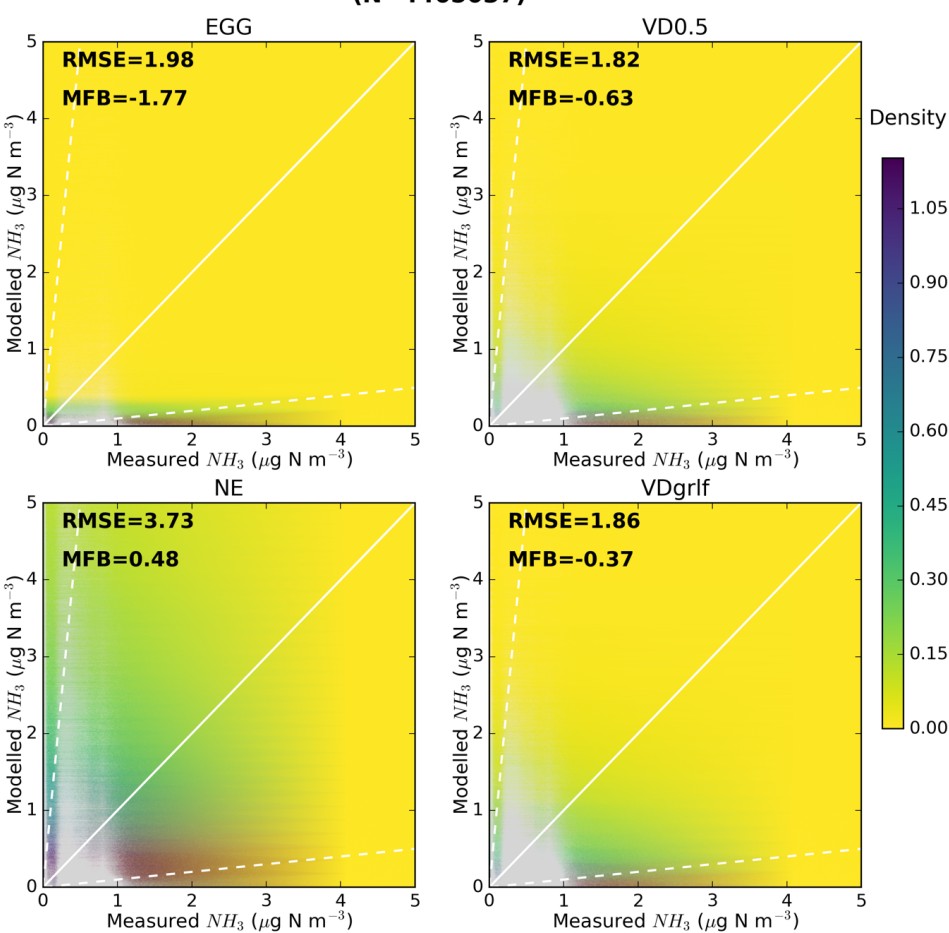

Figure 8. Kernel density estimation (KDE) of the probability density function (PDF) of modelled versus CrIS concentrations of ammonia in a non-parametric way. Modelled concentrations are results of simulations using different emissions datasets (EGG, VD0.5, NE and VDgrlf) for 2012–2017.


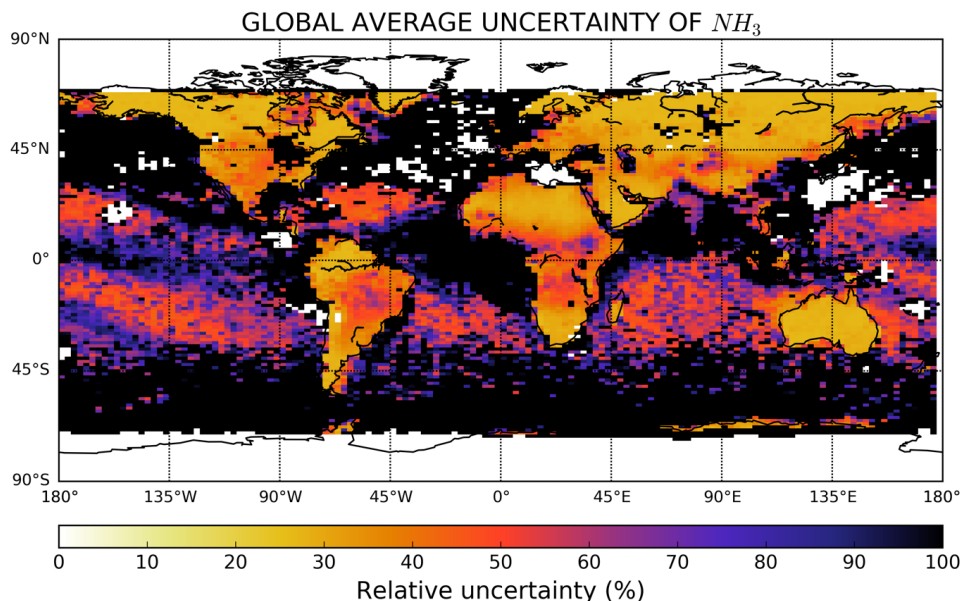

Figure 9. 10-year average relative uncertainty of modelled surface concentrations expressed as the standard deviation of surface concentrations from a model ensemble (Table 1) divided by the average.



Table 1. Model ensemble simulations using different emissions for ammonia that were used in
the calculations of uncertainty. Uncertainties were calculated as the standard deviation of the
surface concentrations of ammonia from the 10 ensemble members for the 10-year period
(2008–2017).

|  | Parameter perturbed | 10-year average emissions (Tg yr$^{-1}$) |
|---|---|---|
| Ensemble 1 | $d_k = 0$ in Eq. 2 | 121±50.6 |
| Ensemble 2 | $d_k = 10$ in Eq. 2 | 175±33.3 |
| Ensemble 3 | $d_k = 20$ in Eq. 2 | 189±28.7 |
| Ensemble 4 | $d_k = 60$ in Eq. 2 | 218±15.5 |
| Ensemble 5 | $d_k = 100$ in Eq. 2 | 208±51.8 |
| Ensemble 6 | $d_k = 500$ in Eq. 2 | 223±26.5 |
| Ensemble 7 | EGG | 65±2.8 |
| Ensemble 8 | VD0.5 | 189 |
| Ensemble 9 | NE | 213±18.1 |
| Ensemble 10 | VDgrlf | 201±10.4 |

