# Peer review of "10-year satellite-constrained fluxes of ammonia improve"

_Atmospheric Chemistry and Physics, 2020_

## Referee Comment (RC1) · Anonymous Referee #1 · 26 Nov 2020

This manuscript provides a description of an inverse method based on the NH3 life-time to estimate NH3 global emissions from the satellite IASI observations over the 10 yr-period 2008-2017. As NH3 is a key species for understanding the PM levels, the quantification of its global emissions is important and would be useful to a wide community. The authors cover an important topic, appropriate for ACP. Nevertheless, I have some major comments listed below that should be considered by the authors before publication.

Major comments:

1/ The fact that NH3 columns in the atmosphere depend not only on NH3 emissions,

but is also linked to the abundance of nitric and sulfuric acids (and consequently to NOx and SO2 emissions) is not fully described. To tackle the large variability of the ammonia lifetime, the authors calculated the NH3 lifetime with a CTM and the spatial variability of ammonia is taken into account. I have more doubt about the temporal variability of ammonia and its main drivers in the atmosphere. If I well understand, the variable lifetime chosen for this study is a gridded average over the 10-yr period. If it is correct, the temporal trend in nitric and sulfuric acids is not fully taken into account, while it could have an importance for the deduced NH3 emissions over a 10-yr period. This choice should be explained in the text. Would it possible to calculate yearly lifetimes as a sensitivity test to assess the robustness of your study?

2/ A comprehensive overview about the existing literature is missing. For example, result for SO2 changes in Figure S2 is not in agreement with Krotkov et al., 2016, ACP, showing strong decrease of SO2 between 2005 and 2015 at least over Eastern US and over Eastern Europe. Also, different publications have shown NH3 peak in spring over northwestern European countries, not seen here. At least, discrepancies with previous studies should be discussed. These features could be explained by the choice of the authors to analyze their results for Europe or for the US as a whole. An analysis done for the hot-spot regions, of interest, where the emissions are high in Figure 4 may help the analysis.

3/ The impact of the abundance of sulfuric acid on NH3 columns is detailed, but not the impact of the abundance of nitric acid. Is this impact considered negligible compared to those of sulfuric acids? This should be discussed. The same Figure S2 for NO2 columns and nitrate concentrations may help analyzing the results.

Specific comments:

line 87: a comma is missing before "the Tropospheric Emission Spectrometer"

line 90-95: a verb is missing in this sentence

line 96-97: Note that Kuenen and Dore, [2019] estimated the uncertainties linked to the agricultural sector at about 100-300% at the European and annual scale. https://www.eea.europa.eu/publications/emep-eea-guidebook-2019/part-a-general-guidance-chapters/5-uncertainties/view

line 98-102 : What is the differences between the different IASI products? The terms NE, VD0.5 and VDgrlf are not intuitive and are not explained at this stage.

line 105: please add references of studies using this state-of-the-art inventory.

Line 124: could a difference of 2%±24% just due to the use of particular vertical profiles be interpreted as "small uncertainties"?

Line 126-151: the description for CrIS gives more information than for IASI. The analysis of the results may be facilitating with the same information for both the instruments. I encourage you to give more information for IASI (total column uncertainties, peak sensitivity, detection limit, etc)

Line 152, Section 2.2: could you please provide a map of the interpolated IASI observations? As you performed simulations, it would be great to see the comparison between IASI and the CTM.

Line 155: What is the CTM? As the variable lifetime in section 2.3 is based on this CTM, it should be described before. I would have described LMDZ-OR-INCA before section 2.3.

Line 160: I would refer to IASI ammonia total columns.

Line 188: Please precise the regions where nitric and sulfuric acids are abundant in the text or at least, refer to Figure 2c and to Figure 2d.

Line 211: Is the variable lifetime from a CTM for the quantification of VDgrlf emissions similar to the one for the quantification of NE emissions? This is not clear.

Line 227-239: Has the NH3 deposition of LMDz-OR-INCA been already evaluated? Is

the bi-directional exchange with surfaces taken into account? This is not discussed. If not, how does it impact your NH3 emissions?

Line 253: you do not focus on hotspot regions but on continents as a whole.

Line 256-271: the different lifetimes of the literature and your results could be highlighted in a Table. Line 276: As Ammonia lifetime depends on the presence of ammonia's reactants (sulfuric and nitric acid), it also depends on NOx and SO2 emissions, not only NH3 emissions. I would have written "(sulfuric and nitric acids, through SO2 and NOx emissions)".

Figure 1: space is missing between the legend and Figure 1c and 1d

Line 287: "which is in the range of the previously reported values". Your results are far from the results from Dammers et al [2019] for example. How do you explain such differences? Could the simulated NH3 lifetime by CTM be over-estimated?

Line 296: Please note in the legend of Figure 1b that the average ammonia emissions are calculated from the 10-year IASI observations and precise with which lifetime. I first thought it was the average ammonia emissions from ECLIPSEv5-GFED4-GEIA. Please also verify the legend of Figure S3.

Line 320-321: The sentence "Although column concentrations of both sulfur dioxide and sulfates present strong interannual variability, they do not show significant changes on an annual basis" is not clear. Please rephrase.

Line 331: I do not understand why the anomaly is calculated only after 2015. Please explain.

Line 334-337: why the NH3 emissions based on IASI observations could be impacted by changes in SO2 and NOx emissions only after 2015? In Lachatre et al., 2019, the study you are citing line 337, the changes in SO2 at least are seen before 2015. This is also the case in your Figure S2. Please strengthen this discussion.

Line 352: please deeply detail why the fact that northern India has been previously identified as a hot-spot region for ammonia explains the differences between the emission datasets.

Line 335: Please verify the species indices

Line 356: the ammonia emissions remain mostly constant at the global scale. Is it still true at continental scale?

Line 357: "The total calculated ammonia emissions": which one?

Line 360-363: could you please provide statistics (average and standard deviation) for South American and European emissions as well as for the global budget?

Line 363-364: "Based upon IASI retrievals, Liu et al. (2019) showed an increase of surface NH3 concentrations trend of more than 0.2$\mu$gNm-3yr-1": I do not understand the link with the previous sentence.

Line 365: "Ammonia emissions derived over China in this work are among the highest worldwide (Figure S1)": is this already the case in the EDGAR and EGG bottom-up inventories or is this a new feature?

Line 370: please precise "The comparison of the annual ammonia NE emissions..." In general, you should specify the inventory or the sensitivity test you are referring to, it would help for the reading and for the understanding of the study.

Line 377: I would add "in these regions" at the end of the sentence. Indeed, the impact of the different lifetimes seems to be slight over the other regions of the world.

Line 385-386: is this contradictory with the sentence "European emissions are practically identical in all datasets" in line 361?

Line 460: consist in?

Line 461-470: The description of the different inventories and of the different performed

simulations should occur before in the text. I would have placed this paragraph at the end of the introduction.

Figure 4: you should number the different graphs. It would be easier to reference them in the text. Please better describe the NH3 emission dataset in the legend.

Line 532: there is an empty bracket.

Section 4.2: Does the evaluation against CrIS done at the global scale? It is not specified. If it is the case, it is not comparable with the surface evaluation done at the regional scale. It would be very interested to do it also at regional scale for the analysis, as in Figure 5, 6 and 7 and particularly over hot-spots as explained in the major comments.

Line 599: the word "already" is misplaced in the sentence.

Figure 9: the colors of the scale should be changed: when the uncertainty is high, the borders on the map are not clearly visible.

Line 612: what are the regions with "changing balance between nitrate and sulfate abundances"? Please detail in the text.
* * *

---

## Author Comment (AC1) · 16 Jan 2021

This manuscript provides a description of an inverse method based on the NH3 life-time to estimate NH3 global emissions from the satellite IASI observations over the 10 yr-period 2008-2017. As NH3 is a key species for understanding the PM levels, the quantification of its global emissions is important and would be useful to a wide community. The authors cover an important topic, appropriate for ACP. Nevertheless, I have some major comments listed below that should be considered by the authors before publication.

Response: We acknowledge reviewer's effort to improve our manuscript.

[Figure]

Major comments:

1/ The fact that NH3 columns in the atmosphere depend not only on NH3 emissions, but is also linked to the abundance of nitric and sulfuric acids (and consequently to NOx and SO2 emissions) is not fully described. To tackle the large variability of the ammonia lifetime, the authors calculated the NH3 lifetime with a CTM and the spatial variability of ammonia is taken into account. I have more doubt about the temporal variability of ammonia and its main drivers in the atmosphere. If I well understand, the variable lifetime chosen for this study is a gridded average over the 10-yr period. If it is correct, the temporal trend in nitric and sulfuric acids is not fully taken into account, while it could have an importance for the deduced NH3 emissions over a 10-yr period. This choice should be explained in the text. Would it possible to calculate yearly lifetimes as a sensitivity test to assess the robustness of your study?

Response: We appreciate reviewer's help to clarify this very important issue. As seen in Figure 1d and explained in the legend, the lifetime, as well as the emissions were calculated in monthly timesteps.

However, we admit this is not clear in the text, and therefore we have tried to clarify it further there. Some examples of our corrections are in section 2.3 (second paragraph, see Track Changes), Section 3 (first paragraph, see Track Changes), section 3.2 (first paragraph, see Track Changes). As we show in Figure 1d, the temporal trends of ammonia's reactants are considered and appear to have an effect on the lifetime, which varies from 10.3 to 12.2 hours.

2/ A comprehensive overview about the existing literature is missing. For example, result for SO2 changes in Figure S2 is not in agreement with Krotkov et al., 2016, ACP, showing strong decrease of SO2 between 2005 and 2015 at least over Eastern US and over Eastern Europe. Also, different publications have shown NH3 peak in spring over northwestern European countries, not seen here. At least, discrepancies with previous studies should be discussed. These features could be explained by the choice of the

authors to analyze their results for Europe or for the US as a whole. An analysis done for the hot-spot regions, of interest, where the emissions are high in Figure 4 may help the analysis.

Response: The legend of SO2 explains that these are not results from our model/set-up, but assimilated data from NASA's OMI (Ozone Monitoring Instrument) and MERRA2 (Modern-Era Retrospective Analysis for Research and Applications, Version 2). This is also explained in the manuscript (section 3.2, third paragraph, see Track Changes). About the seasonal variability of the NH3 emissions, we agree with the reviewer that the spring peaks over northwestern European countries are not seen, because of our choice on the presentation of these results. Since we conduct a global study, we have chosen to study continental emissions rather than focusing only on hot-spot regions. The reason why we did this is because the aim of the paper is not to study the hot-spot emissions of NH3 as seen from IASI. This has been highlighted already by Van Damme et al. Nature paper (see reference list of the manuscript). We focus on how the prescribed emissions retrieved from IASI can improve modelled concentrations and if models need higher emissions to capture measured concentrations. As a response to if our results are consistent with those of northwestern European countries highlighted in other papers, we plot seasonal emissions of NH3 for all years, as in Figure 4 of the manuscript (see Fig. 1 below).

Except for years 2013 and 2015 that peak in summertime, all other years peaked in spring, which is in agreement with the reported hot-spot emissions in northwestern Europe.

3/ The impact of the abundance of sulfuric acid on NH3 columns is detailed, but not the impact of the abundance of nitric acid. Is this impact considered negligible compared to those of sulfuric acids? This should be discussed. The same Figure S2 for NO2 columns and nitrate concentrations may help analyzing the results.

Response: We agree with the reviewer. Reactions with nitric acid are not negligible. However, they may have different results in NH3 concentrations depending on the physicochemical parameters as we explain in the text (neutralization or production of NH3). We have retrieved NO2 from OMI, in consistency with SO2, which we now present in Figure 2 of the manuscript and discuss in the text (section 3.2, circa p. 320-340).

Specific comments:

line 87: a comma is missing before "the Tropospheric Emission Spectrometer"

Response: Corrected (see Track Changes, circa L.87).

line 90-95: a verb is missing in this sentence

Response: Corrected (see Track Changes, circa L.91).

line 96-97: Note that Kuenen and Dore, [2019] estimated the uncertainties linked to the agricultural sector at about 100-300% at the European and annual scale. https://www.eea.europa.eu/publications/emep-eea-guidebook-2019/part-a-general-guidance-chapters/5-uncertainties/view

Response: We have added this useful information in the manuscript (see Track Changes, circa L.98).

line 98-102 : What is the differences between the different IASI products? The terms NE, VD0.5 and VDgrlf are not intuitive and are not explained at this stage.

Response: In principle, we agree with the reviewer here. However, we cannot add methodological details in the Introduction, and we'd rather prefer to leave only the names of the different datasets used in the analysis. Further down in the Methods section, we explain in detail what each name refers to and how the results were obtained. We have added a sentence explaining this in circa L.106 (see Track Changes).

line 105: please add references of studies using this state-of-the-art inventory.

Response: The sentence we have added in circa L.106 (see Track Changes) refers to all emission datasets used in the present study including the state-of-the-art emissions from ECLIPSE-GFED4-GEIA and EDGAR-GFED4.

Line 124: could a difference of 2%±24% just due to the use of particular vertical profiles be interpreted as "small uncertainties"?

Response: We are not sure if we can judge the reported by Van Damme et al. (2018) values on uncertainties. However, as it is stated in their paper, the calculation does not refer to just particular vertical profiles, but rather to a global average: "Differences between columns derived with a fixed vertical profile (baseline) and columns derived using variable modelled profiles are of the order of 2% ± 24% on a global scale, but may be substantially larger for individual locations linked to regional differences in meteorological mixing and recirculation."

Line 126-151: the description for CrIS gives more information than for IASI. The analysis of the results may be facilitating with the same information for both the instruments. I encourage you to give more information for IASI (total column uncertainties, peak sensitivity, detection limit, etc).

Response: We have added further details on errors and detection limits for IASI ammonia (see Track Changes in section 2.1.1). Though, we have tried to keep the length of the section consistent with this of CrIS and avoid repetitions, since detailed information of the product is published elsewhere (see references within the manuscript).

Line 152, Section 2.2: could you please provide a map of the interpolated IASI observations? As you performed simulations, it would be great to see the comparison between IASI and the CTM.

Response: We have added this plot in the Supplementary Figure 11, which gives an example of how the gridded results of IASI ammonia compares to the raw data. We believe it is more appropriate to show it there.

[Figure]

Line 155: What is the CTM? As the variable lifetime in section 2.3 is based on this CTM, it should be described before. I would have described LMDZ-OR-INCA before section 2.3.

Response: We agree with the reviewer that the structure was awkward. We have moved the presentation of the CTM first in section 2 of the Methodology (see Track Changes section 2).

Line 160: I would refer to IASI ammonia total columns.

Response: Corrected. Please check at circa L. 214 (Track Changes).

Line 188: Please precise the regions where nitric and sulfuric acids are abundant in the text or at least, refer to Figure 2c and to Figure 2d.

Response: At this point, we discuss the method in general and do not refer to our results. We say that the use of a variable lifetime, and not a constant one, will be able to capture any variability caused be the chemical reactions of ammonia in the atmosphere, where and if they occur.

Line 211: Is the variable lifetime from a CTM for the quantification of VDgrlf emissions similar to the one for the quantification of NE emissions? This is not clear.

Response: We appreciate reviewer's help here. Indeed, this is not clear, and we have now corrected this part (see Track Changes at circa L. 266 of section 2.4).

Line 227-239: Has the NH3 deposition of LMDz-OR-INCA been already evaluated? Is the bi-directional exchange with surfaces taken into account? This is not discussed. If not, how does it impact your NH3 emissions?

Response: The total deposition of SOx, (SO2+SO42+), NHx (NH3+NH4+), and NOy (NO+NO2+NO3+HNO2+HNO3+HNO4+N2O5+organic nitrates+particulate NO3-) have been evaluated (see Hauglustaine et al., 2014, in the manuscript). However, we admit we do not account for a compensation parameterization in the CTM, as highlighted by the reviewer. We only have the emissions on one side, and the dry deposition ion on the other.

Line 253: you do not focus on hotspot regions but on continents as a whole.

Response: This is true; we agree with the reviewer and we have amended the text at this point (see Track Changes in circa L. 276, first paragraph of section 3).

Line 256-271: the different lifetimes of the literature and your results could be highlighted in a Table.

Response: There is a relevant supplementary Table in Van Damme et al. (2018) Nature paper (see reference within the manuscript), which presents literature values for ammonia lifetimes. We point to this table as " The atmospheric lifetimes of ammonia were summarized in Van Damme et al. (2018)." We do not want to be repetitive and put the same Table here. If the reviewer/editor has a different suggestion, we are willing to correct this in a next stage.

Line 276: As Ammonia lifetime depends on the presence of ammonia's reactants (sulfuric and nitric acid), it also depends on NOx and SO2 emissions, not only NH3 emissions. I would have written "(sulfuric and nitric acids, through SO2 and NOx emissions)".

Response: We agree with the reviewer. As we have now clarified in the text, NO2 and SO2 are precursors of ammonias atmospheric reactants, hence lifetime is indirectly linked to their concentrations. We have followed reviewer's suggestion to amend this sentence (see Track Changes in circa L. 376, p.10).

Figure 1: space is missing between the legend and Figure 1c and 1d

Response: We are not sure we understand where the problem is in Figure 1. Both the legend and the figure appear to be fine in our version. We have corrected some space problems in the title of Fig. 1c (reactants of NH3) that were overplotted by latitudinal values. If the reviewer still thinks there's a space missing somewhere, we could correct

it in a next stage of the reviewing process.

Line 287: "which is in the range of the previously reported values". Your results are far from the results from Dammers et al [2019] for example. How do you explain such differences? Could the simulated NH3 lifetime by CTM be over-estimated?

Response: We cannot judge the values calculated by Dammers et al. [2019]. As we report in circa L. 367-370 "The majority of ammonia lifetimes reported regionally or globally fall within 10 and 24 hours independently of the different approaches (Hauglustaine et al., 2014; Hertel et al., 2012; Möller and Schieferdecker, 1985; Sutton et al., 1993; Whitburn et al., 2016b),. . .".

Line 296: Please note in the legend of Figure 1b that the average ammonia emissions are calculated from the 10-year IASI observations and precise with which lifetime. I first thought it was the average ammonia emissions from ECLIPSEv5-GFED4-GEIA. Please also verify the legend of Figure S3.

Response: We thank the reviewer here; We have now clarified that the plot refers to the NE emissions (Track Changes at legend of Figure 1). We have also clarified this in the legend of the supplementary Figure S3.

Line 320-321: The sentence "Although column concentrations of both sulfur dioxide and sulfates present strong interannual variability, they do not show significant changes on an annual basis" is not clear. Please rephrase.

Response: We have amended this sentence to be consistent with what the figures show. Please see Track Changes at circa L. 428-432 (p.12).

Line 331: I do not understand why the anomaly is calculated only after 2015. Please explain.

Response: We initially thought to study anomalies after 2015, as our calculated emissions seem to increase after 2015. We agree with the reviewer that changes are already obvious since 2012 and now provide a more complete reasoning supported by

relevant references. However, the largest reductions were seen after 2015, in agreement with the emissions of NH3 that we present here, as seen in the attached Fig.2 and that is why we have chosen to restrict anomalies after 2015. Please see Track Changes at p.11-12.

Line 334-337: why the NH3 emissions based on IASI observations could be impacted by changes in SO2 and NOx emissions only after 2015? In Lachatre et al., 2019, the study you are citing line 337, the changes in SO2 at least are seen before 2015. This is also the case in your Figure S2. Please strengthen this discussion.

Response: As we now discuss in L. 418-p.11, although the SO2 and NOx reduction is evident since 2012, the largest changes are calculated for the period after 2015, which is in agreement with our suggested NH3 emissions. This is also evident if we compare anomalies after 2012 with those after 2015 as in the attached Fig.2. Therefore, we present anomalies after 2015. We have tried to explain this in the manuscript (please see Track Changes in p.11-12).

Line 352: please deeply detail why the fact that northern India has been previously identified as a hot-spot region for ammonia explains the differences between the emission datasets.

Response: We believe we do not imply that the fact that N. India has been identified as a hotspot region explains the difference in the emission datasets. We only say that these hotspot emissions in N. India have been highlighted ti be due to agricultural activities and we give 2 references to support this. We have now tried to re-write the sentence (see Track Changes at circa L. 476).

Line 335: Please verify the species indices

Response: We have amended this part and the overall discussion in this section as explained in previous comments (see Track Changes at p. 11-12).

Line 356: the ammonia emissions remain mostly constant at the global scale. Is it still

true at continental scale?

Response: Yes, it is actually true that no significant continental changes occurred. For example, the ECLIPSE emissions which are based on the GAINS model are produced for 5-year timesteps. What global models assume is usually a linear interpolation to scale the emissions for each of the years in between. For justification, we plot the annual emissions from ECLIPSEv5-GFED4-GEIA and from EDGARv4.3.1-GFED4 in the attached Fig. 3 and 4.

Line 357: "The total calculated ammonia emissions": which one?

Response: We have amended this sentence (Track Changes at circa L.641-642).

Line 360-363: could you please provide statistics (average and standard deviation) for South American and European emissions as well as for the global budget?

Response: We have amended this part. Numbers have been added everywhere in this paragraph presenting average and sd (see Track Changes at L.491-496, p. 13).

Line 363-364: "Based upon IASI retrievals, Liu et al. (2019) showed an increase of surface NH3 concentrations trend of more than $0.2\mu gNm-3yr-1$": I do not understand the link with the previous sentence.

Response: We consent with this comment and we have removed this sentence thanks to the reviewer (see Track Changes at circa L.496, p. 13).

Line 365: "Ammonia emissions derived over China in this work are among the highest worldwide (Figure S1)": is this already the case in the EDGAR and EGG bottom-up inventories or is this a new feature?

Response: We have made clear that by saying "in this work" we mean the emissions highlighted as NE (see Track Changes at circa L.496, p. 13).

Line 370: please precise "The comparison of the annual ammonia NE emissions. . ." In general, you should specify the inventory or the sensitivity test you are referring to,

it would help for the reading and for the understanding of the study.

Response: The reviewer is again right here. We have modified the sentence as follows: "The comparison of the annual ammonia emissions in the NE dataset to the . . ." (see Track Changes, L.509, p.14).

Line 377: I would add "in these regions" at the end of the sentence. Indeed, the impact of the different lifetimes seems to be slight over the other regions of the world.

Response: Corrected as suggested by the reviewer (see Track Changes, L.517, p.14)

Line 385-386: is this contradictory with the sentence "European emissions are practically identical in all datasets" in line 361?

Response: We have modified the sentence as follows ". . . in all datasets except EGG . . ." (see Track Changes, L.493, p.13).

Line 460: consist in?

Response: We have corrected as suggested (see Track Changes, L.601, p.17).

Line 461-470: The description of the different inventories and of the different performed simulations should occur before in the text. I would have placed this paragraph at the end of the introduction.

Response: The sequence of the paper is (a) proof that modelled lifetimes are realistic, (b) presentations of the different emission inventories for NH3 based on different methodologies, (c) comparison with state of the art datasets (ECLIPSE-GFED-GEIA, EDGAR-GFED) that are frequently used to simulate NH3 concentrations in global models. Then, we need to prove that the emissions presented in the paper produced more realistic modelled concentrations, and for this reason, we simulate NH3 using each of the different emissions and compare model concentrations with surface measurements and satellite data. We explain all these in an introductory paragraph in discussions. However, we agree with the reviewer and have moved the part that explains what the

EGG emissions refer to into the place that appear for the first time. Instead, we use abbreviations everywhere in this paragraph.

Figure 4: you should number the different graphs. It would be easier to reference them in the text. Please better describe the NH3 emission dataset in the legend.

Response: We do not really use the numbering in any part of the text when refer to this figure. This is mainly done because each graph placed in any raw shows exactly the same thing for different continental regions. We do not think this is necessary (since it's not used) and if the reviewer/editor insists, we may do so in a next step.

Line 532: there is an empty bracket.

Response: Bracket has now been removed (see Track Changes, p.19).

Section 4.2: Does the evaluation against CrIS done at the global scale? It is not specified. If it is the case, it is not comparable with the surface evaluation done at the regional scale. It would be very interested to do it also at regional scale for the analysis, as in Figure 5, 6 and 7 and particularly over hot-spots as explained in the major comments.

Response: Yes, the comparison with CrIS NH3 refers to global data, which we now specify in L. 677-678. As we already answered in a previous comment, an evaluation of IASI ammonia for several hotspot regions has been done in Van Damme et al. (2018) Nature paper. What we do here is to use IASI NH3 to produce emissions and see if a model that participates in CMIP and IPCC simulations can improve its performance, also giving these emissions to be used by anyone interested. We evaluate the modelled concentrations against ground measurements that we trust more, in general for N. America, Europe and Southeastern Asia. As a supplement we compare with another global product (CrIS), to prove that concentrations are better reproduced, not only in N. America, Europe and Southeastern Asia, but in a global scale.

Line 599: the word "already" is misplaced in the sentence.

Response: "already" should be "although" in this sentence. We thank the reviewer for pointing this out. We have amended the sentence (see Track Changes, L.736, p.21).

Figure 9: the colors of the scale should be changed: when the uncertainty is high, the borders on the map are not clearly visible.

Response: We have used another colormap as suggested by the reviewer, in order to have visible coastlines (see manuscript with Track Changes).

Line 612: what are the regions with "changing balance between nitrate and sulfate abundances"? Please detail in the text.

Response: We have amended the sentence to make a more concrete statement as suggested (see Track Changes as L. 751-753, p. 21).
* * *
[Figure]

**Northwestern European countries**

[Figure]

**Fig. 1.** Seasonal emissions of NH3 in northwestern European countries.

[Figure]

**Fig. 2.** Left column: SO2, NO2 and SO4 anomalies after 2015, as in the manuscript. Right column: same anomalies calculated after 2012, when the first reductions of these precursors were observed.

[Figure]

**Fig. 3.** Annual emissions of NH3 in ECLIPSEv5-GFED4-GEIA.

EDGARv4.3.1-GFED4

[Figure]

**Fig. 4.** Annual emissions of NH3 in EDGARv4.3.1-GFED4.

---

## Author Comment (AC2) · 16 Jan 2021

- Overall, the paper is well written and provides new information to the literature on global NH3, which has not been well characterized previously. The paper is rather long and could condense it down to a tighter paper that is more focused on key results and conclusions.

Response: We appreciate reviewer's comments and his willingness to improve this manuscript. We have made all the changes requested by the 2 reviewers and we are willing to further work to shorten the manuscript, if additional detailed comments are to be requested.

[Figure]

- Comparing model predictions at the coarse level presented here (2.5 degrees or 250 km) to ground monitors and discussing "hotspots" may not well represent the spatially variable nature of NH3 emissions. Averages over these large cells could misrepresent key features of NH3 distributions. However, the spatial resolution in Figure 4 appears to be finer than 2.5 x 1.3 degree. Was a particular plotting technique used to show the NH3 levels that might be making gradient interpolations or is the data in Figure 4 actually 2.5 x 1.3 degree resolution?

Response: We acknowledge reviewer's observation here and we admit this was misleading. We have now clarified in section 2.3 (see Track Changes at p.7-8). What we have done was to process the IASI column concentration measurements onto a grid of $0.5°×0.5°$ using the IDW method that we describe in section 2.3. Then, since the resolution of the CTM model that we used bilinear interpolation classic method to convert to the model resolution $(2.5°×1.3°)$.

- The authors estimated emission fluxes using a lifetime parameter from the CTM. Was there some reason a traditional assimilation approach (e.g., like Alvarado is doing) was not included in this assessment? The authors should consider a comparison of the column predictions of the CTM simulation using the estimated emissions back to the IASI measurements. If the lifetime approach is accurate, the CTM should accurately predict the IASI columns when using the scaled emissions. If this was done it is not clear from the text. Further, it did not seem like the seasonal NH3 lifetime estimated by the CTM provided a substantively different result than the 0.5VD constant assumption.

Response: There was not special reason for not using a classic assimilation method here. Our idea was to try to calculate emissions from IASI column NH3 measurements. For this, we needed a metric of the lifetime of NH3. We used a constant lifetime of 0.5 d everywhere, as well as a gridded one, calculated from a model, which we thought it is more realistic, as ammonia cannot have the same lifetime everywhere (see section 3.1). Finally, we wanted to see if the calculated emissions have a significant impact on surface concentrations. For this, we compare with measurements from EMEP, EANET

and AMoN. The goal of this paper is not to validate the CTM against IASI column ammonia. However, an example of how the column NH3 in the model compares to IASI column NH3 is now given in Supplementary Figure S11 (of the manuscript). The model is continuously validated by the LSCE group (see relevant papers here: https://www.lsce.ipsl.fr/en/Phocea/Vie_des_labos/Ast/ast_groupe.php?id_groupe=94&voir=publis). We rather want to prove that for very short-lived species such as NH3, a simple approach like the one we describe in section 2 is enough to constrain the main source of NH3 in the atmosphere, a chemical species that is difficult to be quantified with classic inverse modelling approaches, due to its heterogeneous chemistry. The goals of this manuscript are explained in detail in the last paragraph of the introduction.

The reviewer states that ", it did not seem like the seasonal NH3 lifetime estimated by the CTM provided a substantively different result than the 0.5VD constant assumption". We only show the lifetime calculated by the CTM in Figure 1d (of the manuscript), which basically shows values between 10 – 12.5 hours, whereas in VD0.5 a constant lifetime of 12 hours (0.5 d) was used everywhere. The difference in the emissions using a variable versus a constant lifetime for NH3 are shown in Figure 3 (of the manuscript) and they are as high as 29.4 Tg/y (on average), or 15% different, which we do think it is substantially different; both in absolute numbers, but also in the spatial distribution of the emissions. The impact on the surface concentrations against observations is shown in both as time-series plots in the Supplements and as scatterplots in the main text. The IASI-constrained emissions, at least in the North America and Southeastern Asia, capture realistically atmospheric concentrations (see linear scale in x- and y-axes).

- Ammonia has a strong diurnal profile. Does the assumption for diel profile impact any of the results presented in this paper or does the diurnal nature of NH3 emissions have no impact on these products?

Response: Indeed, NH3 has a strong diurnal cycle, and the CTM uses a means to account for a diurnal cycle. However, we have not assessed how the diurnal cycle

in the present setup affects the results. The reason is that, although IASI NH3 are measured twice a day, only morning measurements were used in the present study, due to the larger thermal conditions that lead to smaller uncertainties. Accordingly, we have used daily model outputs for concentrations and monthly mean lifetimes from the model. In addition, all measurements used here to evaluate modelled concentrations have a temporal resolution of 1 or 2 weeks. Therefore, no further effort to deal with the diurnal cycle of NH3 was made and rather assumed that it should not affect much our results. Of course, we have to admit that a bias in the overall assessment could be realistic, although no data to prove this were available.

- When taking a closer look at Figure 6, is it surprising that VD0.5, NE, and VDgrid emisisons used in a model result in very few model estimations of ammonia below 0.5 and ECG rarely has a prediction above 0.5. Some of the calculated performance metrics may suggest "good" model performance but the shape of the model-observed NH3 in Figure 6 shows some features that suggest they many of these approaches can not replicate the range of NH3 levels measured.

Response: We rather think this is normal. As one can read in section 3.4, "North American annual ammonia emissions over the 10-year period were averaged 1.1+-0.1 Tg yr-1 (average+-sd). These values are over two orders of magnitude higher than those in EGG (0.062+-0.0013 Tg yr-1). Note that his estimate is three times lower than those reported in VD0.5 (3.1 Tg yr-1) or in VDgrlf (3.4+-0.5 Tg yr-1)." Therefore, we see smaller MFB values (=0.32) in Figure 6 (of the manuscript) than those of VD0.5 (=0.52) and VDgrlf (=0.54) and much higher than those in EGG (=-0.28). Another view of the modelled-observation mismatches can be seen in supplementary Figures S7-S9.

- Please provide some more clarity on the vertical profile used for NH3 for IASI retrievals. Is this constant and not variable with changes in altitude? Does the vertical profile conform to profiles measured as part of aircraft measurement campaigns and seem realistic?

Response: We have not used any vertical profile for IASI NH3. As we explain in detail in p.7-L.316 "IASI total column ammonia measurements were interpolated onto . . ." a grid with the method described in section 2.3. Then, a box-model was used to calculate gridded emissions of NH3, as described in section 2.4 (L. 368-370, p. 8): "It takes into account the gridded column concentrations of ammonia that were calculated with the IDW interpolation method and all the potential removal processes of ammonia occurring in a hypothetical atmospheric box. . .".

- Line 581: What are large sources of anthropogenic NH3 in central USA?

Response: We explain this, two paragraphs before this point. Please check manuscript with Track Changes (l.599-618): "First, a small region in Colorado, Central US, which is the location of a large agricultural region that traditionally releases large ammonia emissions. . .". Then, we continue explaining main sources in Central US "is the state of Iowa (home to more than 20 million swine, 54 million chickens, and 4 million cattle), northern Texas and Kansas (beef cattle) . . .". We think it is a repetition to mention again and again something that has been explained a few lines before.

- Figure 8 is very hard to interpret. The authors should consider alternative colors or another way to present these results.

Response: We have chosen to use the Gaussian kernel density estimation (KDE) method due to the large amount of data that we had to process, and we thought we should avoid overplotting. Another way to show the improvement of the results would be simple scatterplots that present annual data from all 4 simulations (Fig. 1). The reviewer/editor can possibly decide which one shows better. We would rather prefer the KDE method.
* * *
[Figure]

[Figure]

**Fig. 1.** Annual scatterplots of modelled versus CrIS NH3 surface concentrations from the relevant four simulations using emissions from EGG, VD0.5, NE and VDgrlf.

---

## Author Response (AR2)

Editor Decision: Publish subject to minor revisions (review by editor) (04 Feb 2021) by
Drew Gentner

Comments to the Author:

Please fully consider and address the comments in the recent re-review (copied here for your convenience):

This version of the manuscript has been improved relative to original submission, addressing several of previous reviewers comments. I commend the authors for this revision. However, while this is relevant to the scope of ACP, I have still some concerns with regards to the analysis of the results and to the evaluation of the updated emissions. I suggest minor revisions to address the following concerns before publication:

**Response:** We have addressed all the comments requested. We really agree with the idea to make a more targeted comparison with measurements and satellite data and we now believe that the manuscript has now improved substantially. Please note that we have also shortened the abstract and the conclusions, as there were details that were probably repetitive.

General comments

1/ I note that a particular effort has been done on the evaluation of the results but I still think that the evaluation section could be reinforced. The easiest way to see the effectiveness of the inversion is to show a gridded bias between the observations and the both the simulations using EGG or NE emissions. In this sense, you should complete the Figure S11. In addition, in my opinion, the evaluation against CrIS satellite data should be done at the regional scale in addition to the global one. It would not be an evaluation of the IASI data or of the model but of your updated emissions. It would complete the analysis of the evaluation against surface stations at least over Europe and over the US where your inversion can degrade the fit to the measurements compared to the prior EGG inventory.

**Response**: We really appreciate for this comment, as we had not thought about this kind of presentation previously. Since it is a very valuable figure, we have added the mapped station bias not in the supplements, but as Figure 8 of the main manuscript (see Track Changes). We have also added a paragraph where we try to explain what the main conclusion from this map is (see lines 572-580 in Track Changes).

In addition, we also agree that the comparison with the CrIS data should not refer to all datasets but only to the one we present here (NE). For this reason, we have restricted the comparison only for the NE emissions and only for North America, Europe and Southeastern Asia, as suggested by the Editor. For this we have substituted the previous figure with Figure 9 (see manuscript with Track Changes). Since section 4.2 became really small after limiting the comparison to the NE emission data only, we merged it with section 4.1, which is now entitled "Validation against ground-based observations and satellite products"(see manuscript with Track Changes).

2/ A comprehensive overview about the existing literature is still missing for the analysis of SO2 changes. Figure S2 is not in agreement with Krotkov et al., 2016, ACP, showing strong decrease of SO2 between 2005 and 2015 at least over Eastern US and over Eastern Europe with the OMI data themselves. These features could be explained by the choice of the authors to analyze their results for Europe or for the

US as a whole but it should be discussed in the text. The section 3.2, circa p.320-340 would be more complete with analysis not only for the North China plain, even if I understand the particular interest for this region.

**Response**: We have tried to clarify that the observed decrease after 2015 is only due to our choice to present global averaged, whereas others (Krotkov et al.) have already seen decreased concentrations after 2015 (Track Changes line 369-375). As already mentioned, the reason why we chose North China Plain to investigate changes in NH3, reactants and precursor species is because the largest anomalies of SO2 and NO2 were seen there. We believe it would be beyond the scope of this paper to investigate in detail more regions in this respect, as the only reason for examining SO2 and NO2 was to explain any changes of ammonia's reactants, which would explain change in NH3 emissions. We believe that Krotkov et al. has done a very complete analysis on SO2 and NO2 from OMI, much more detailed than what we have done in the present.

3/ The fact that the bi-directional exchange with surfaces is not taken into account should be mentioned. The potential impact on the inversion results should also appear in the text.

**Response**: Dear Editor, please do correct me if I did not understand well this query. If you mean that we do not discuss in the text the effect of the exchange of ammonia in/out from the hypothetical box (due to atmospheric transport) that we used as a proxy to calculate emissions, then we do not agree. We have dedicated a full section (4.3 Limitations of the present study), where we explicitly state that the exchange due to transport is an issue of the selected methodology and that we believe it is fair to assume it negligible due to the very short lifetime of NH3 in the atmosphere. In case you meant something else, please be more specific and we will act on the manuscript to correct any issue.

Specific comments
Line 329: "To" instead of "to"
**Response**: A full stop was missing there, as the sentence stops before "To". It has been corrected (see Track Changes).

Line 330: "NOx" instead of "NOx"
**Response**: We have now added a subscript at this point (see Track Changes).

Line 329-330: The sentence is not clear. Please rephrase"
**Response**: We now think the sentence is clear after adding the missing full stop. If not, please help us improve this point further.

---

## Author Response (AR3)

Editor's clarification: Note that the reviewer is also requesting a more comprehensive comparison to existing literature in this comment ("A comprehensive overview about the existing literature is still missing for the analysis of SO2 changes.") and is referring Krotkov et al. as an example related to Figure S2. I saw you added a short paragraph on Krotkov et al., but please make sure you have also included comparisons to any other appropriate references to put your results in context.

**Response**: Although I only used SO2 here to investigate if emission changes in Southeastern Asia might have affected satellite NH3, I have done my best to discuss SO2 in more detail adding numerous references that show trends during the last 30 years. Please see Track Changes in Section 3.2.

Editor clarification: It seems to me that the reviewer is asking you to clearly state that bi-directional exchange of NH3 with the Earth surface was not considered. I don't think they are referring to atmospheric transport. Note, they also ask that "The potential impact on the inversion results should also appear in the text." Please resolve the issue and/or respond as appropriate.

**Response**: OK, I now understand. I remember I replied that we do not account for compensation point of NH3 from stomata and tree canopies, but I probably forgot to discuss it in the text. I have now added a few sentences in section 4.3 explaining how this affects surface NH3 in the comparisons, exactly as the reviewer asked for. Please see Track Changes in Section 3.2.